# Alteration in tyrosine phosphorylation of cardiac proteome and EGFR pathway contribute to hypertrophic cardiomyopathy

Mingguo Xu[1,2,13], Kevin C. Bermea [1,13], Marzieh Ayati[3,13], Han Byeol Kim[4], Xiaomei Yang[5], Andres Medina[6], Zongming Fu[7], Amir Heravi[1], Xinyu Zhang[8], Chan Hyun Na [4,9], Allen D. Everett[1], Kathleen Gabrielson[10], D. Brian Foster [11], Nazareno Paolocci[11,12], Anne M. Murphy[1✉] & Genaro A. Ramirez-Correa [1,6✉]

Alterations of serine/threonine phosphorylation of the cardiac proteome are a hallmark of heart failure. However, the contribution of tyrosine phosphorylation (pTyr) to the pathogenesis of cardiac hypertrophy remains unclear. We use global mapping to discover and quantify site-specific pTyr in two cardiac hypertrophic mouse models, i.e., cardiac over-expression of *ErbB2* (TgErbB2) and α myosin heavy chain R403Q (R403Q-αMyHC Tg), compared to control hearts. From this, there are significant phosphoproteomic alterations in TgErbB2 mice in right ventricular cardiomyopathy, hypertrophic cardiomyopathy (HCM), and dilated cardiomyopathy (DCM) pathways. On the other hand, R403Q-αMyHC Tg mice indicated that the EGFR1 pathway is central for cardiac hypertrophy, along with angiopoietin, ErbB, growth hormone, and chemokine signaling pathways activation. Surprisingly, most myofilament proteins have downregulation of pTyr rather than upregulation. Kinase-substrate enrichment analysis (KSEA) shows a marked downregulation of MAPK pathway activity downstream of *k-Ras* in TgErbB2 mice and activation of EGFR, focal adhesion, PDGFR, and actin cytoskeleton pathways. In vivo ErbB2 inhibition by AG-825 decreases cardiomyocyte disarray. Serine/threonine and tyrosine phosphoproteome confirm the above-described pathways and the effectiveness of AG-825 Treatment. Thus, altered pTyr may play a regulatory role in cardiac hypertrophic models.

[1] Department of Pediatrics/Division of Cardiology, Johns Hopkins University School of Medicine, Baltimore, MD, USA. [2] Department of Pediatrics, The Third People's Hospital of Longgang District, Shenzhen 518115, China. [3] Deparment of Computer Science/College of Engineering and Computer Science, University of Texas Rio Grande Valley School of Medicine, Edinburgh, Texas, USA. [4] Department of Neurology/Institute for Cell Engineering, Johns Hopkins University School of Medicine, Baltimore, MD, USA. [5] Department of Anesthesiology, Qilu Hospital, Cheeloo College of Medicine, Shandong University, Ji'nan, China. [6] Department of Molecular Science/UT Health Rio Grande Valley, McAllen, TX, USA. [7] Department of Pediatrics/Division of Hematology, Johns Hopkins University School of Medicine, Baltimore, MD, USA. [8] Department of Cardiology, Qilu Hospital, Cheeloo College of Medicine, Shandong University, Ji'nan, China. [9] Department of Biological Chemistry/McKusick-Nathans Institute of Genetic Medicine, Johns Hopkins University School of Medicine, Baltimore, MD, USA. [10] Department of Molecular and Comparative Pathobiology, Johns Hopkins University School of Medicine, Baltimore, MD, USA. [11] Department of Medicine/Division of Cardiology, Johns Hopkins University School of Medicine, Baltimore, MD, USA. [12] Department of Biomedical Sciences, University of Padova, Padova, Italy. [13] These authors contributed equally: Mingguo Xu, Kevin C. Bermea, Marzieh Ayati. ✉email: murphy@jhmi.edu; genaro.ramirezcorrea@utrgv.edu

Familial hypertrophic cardiomyopathy (HCM) increases the left ventricle (LV) wall thickness; abnormal loading conditions cannot explain this alteration. Mutations in genes encoding sarcomere proteins are common in HCM patients (40–60%)[1]. Studying post-translational modifications (PTMs) of the sarcomere proteins (or $Ca^{2+}$ handling proteins) can offer a unique opportunity to understand better how genetic disorders lead to cardiac dysfunction and discover potential targets for therapy[2–4]. For instance, PTMs of cardiac Troponin I (cTnI), a sarcomere protein centrally involved in myocardial contractility regulation, have been extensively studied, particularly for the functional role of protein kinase A-dependent phosphorylation[5]. Notably, phosphorylation of cTnI-S22/23—one of the most relevant regulatory sites of cTnI—is downregulated in human heart failure (HF)[6] and leads to contractile dysfunction[7].

Tyrosine phosphorylation (pTyr) is essential for cardiac structural development and myofibril organization during embryogenesis[8,9]. For example, several tyrosine phosphatases have been linked to heart disease and even proposed as a therapeutic target for some conditions. Moreover, mutations of the tyrosine-protein phosphatase non-receptor type 11 (PTPN11) can lead to HCM or dilated cardiomyopathy (DCM)[10,11]. In the heart, PTPN11 likely plays a role in systolic dysfunction produced by pressure-overload[12]. Acid phosphatase 1 (ACP1) is another tyrosine phosphatase associated with cardiac pathophysiology. Accordingly, its deletion protects against stress-induced cardiomyopathy[13]. Our group first showed that cTnI-Y26 phosphorylation is readily detected in healthy human hearts and downregulated in human HF and DCM[14]. However, how these alterations contribute to the onset and progression of cardiac disease remains poorly understood. A better grasp of additional site-specific changes of individual tyrosine-phosphorylated sites would help in this direction substantially.

The present study applied a label-free and tandem mass tagging (TMT) quantitative global phosphotyrosine proteomics approach to determine which sarcomere sites have altered amounts of Tyr phosphorylation at specific sites in two unrelated models of HCM. The first model is secondary to the overexpression of the tyrosine kinase receptor ErbB2[15,16]. The second recapitulates features of the human disease, more specifically, an R403Q mutation of the myofilament protein myosin heavy chain, distinctive of familial HCM[17]. TgErbB2 mice initially develop a striking, concentric cardiac hypertrophy, which evolves to diffuse fibrosis and myocyte disarray[16] with HCM. This line of mice also has abnormal calcium handling, is prone to arrhythmias, and to developing hypertrophic obstructive cardiomyopathy[2]. Similarly, R403Q-αMyHC mice reproduce human familial HCM by progressing from mild hypertrophy and fibrosis to overt myocyte disarray, HF, and arrhythmias[17].

The main impetus behind this study was to determine whether triggering HCM through different mechanisms elicits similar pTyr-related pathways/regulatory sites within the heart. In doing so, the manipulation of these pTyr sites would, in turn, offer new opportunities for therapeutic targeting in different forms of human HCM

## Results

### Immunoblotting reveals decreased cardiac pTyr in HCM models.
First, we performed immunoblot analysis of the myocardium of TgErbB2, R403Q-αMyHC, and non-transgenic (Ntg) mice to estimate pTyr. There was a global decline in pTyr in whole heart homogenates of TgErbB2 ($p = 0.0298$) and R403Q-αMyHC ($p = 0.003$) mice compared to Ntg (Fig. 1a, b). The pTyr signal was normalized to troponin I expression levels (Fig. 1a). Although proto-oncogene tyrosine-protein kinase Src (c-Src) is

downstream of ErbB2 signaling, we did not anticipate its increased expression in R403Q-αMyHC mice (Fig. 1c, d, $p = 0.0414$) when normalized to GAPDH western blot signals. The c-Src activity (c-Src Tyr416 phosphorylation) appeared to be enhanced in R403Q-αMyHC mice compared to Ntg; however, it did not reach statistical significance (Fig. 1e, f, one-way ANOVA was used to compare groups). Altogether these data suggest that pTyr is a broadly distributed PTMs in the myocardium. Hence, its upregulation or downregulation may play a regulatory role in the disease progression of non-ischemic cardiomyopathies, as in the case of familial cardiomyopathy (HCM) and DCM, as part of the pathophysiologic response to the underlying disease.

### pTyr in myofilaments and cross-talk with serine/threonine kinases/phosphatases.
Next, we adopted a global TMT phosphotyrosine proteomic approach to compare the whole heart pTyr profile of Ntg, TgErbB2, and R403Q-αMyHC mice (Fig. 2). We hypothesized that unbiased global whole heart pTyr profiling would provide a complete landscape about specific tyrosine sites in essential cardiac-specific proteins and tyrosine kinases, thus providing clues about which tyrosine kinases mediate cardiac pTyr. A total of 1,800 peptide spectra matched (PSM) were collected from the heart ventricle whole proteome, and 50% were identified with high confidence resulting in 213 (R403Q-αMyHC), 214 (Ntg), 217 (TgErbB2) unique tyrosine-phosphorylated peptides corresponding to 499 proteins (Fig. 2g). Tables with all identified phosphoproteins and phosphopeptides are provided in Supplementary Data 3. The mass spectrometry data quality, intensity, and distribution post-median sweep normalization are included in Supplementary Fig. 1a, b. Confidence of phosphorylated site localization was evaluated for annotation, and a score of more than 49% was required. This dataset indicates a comparable yield of peptide identification was achieved, reproducibility on the enrichment of pTyr peptides, and high-quality MS/MS data using similar approaches to others[18–20].

The global TMT phospho-tyrosine proteomics approach identified 499 different tyrosine-phosphorylated proteins, 294 tyrosine-phosphorylated proteins that overlap among the Ntg, and the myocardium from two cardiac hypertrophic models (TgErbB2 and R403Q-αMyHC). At the peptide level, 217 pTyr sites were detected with an overlap of 178 tyrosine-phosphorylated residues (Fig. 2g, h).

Next, the focus was on the cardiac-specific proteins from the myofilament apparatus. Nine major myofilament proteins were tyrosine-phosphorylated. They harbor multiple pTyr amino acid sites, many of which are novel (See Supplementary Fig. 2); the best example is Titin, with 36 pTyr amino acid sites (See Supplementary Fig. 3). Also, seventeen tyrosine kinases had detectable tyrosine-phosphorylated peptides; thus, they could potentially be involved in regulating cardiac myofilament pTyr.

In contrast, only two tyrosine phosphatases (Ptpn11 and Ptpra) had detectable tyrosine-phosphorylated peptides; therefore, they are likely to play a role in regulating cardiac pTyr levels in the sarcomere. In addition to the number of tyrosine kinases found with phosphorylated peptides, fifteen serine/threonine kinases had detectable tyrosine-phosphorylated peptides; however, only one serine/threonine phosphatase (Ppp1r12b) demonstrated tyrosine-phosphorylated peptides. These data show that pTyr is broadly distributed in the heart proteome (cardiac sarcomere and other sub-proteomes) and that a cross-talk between tyrosine kinases/phosphatases with serine/threonine kinases/phosphatases might be regulating cardiac sarcomere pTyr.

### ErbB2 cardiac overexpression and R403Q-αMyHC point mutation remodeled the cardiac pTyr proteome.
Next, we

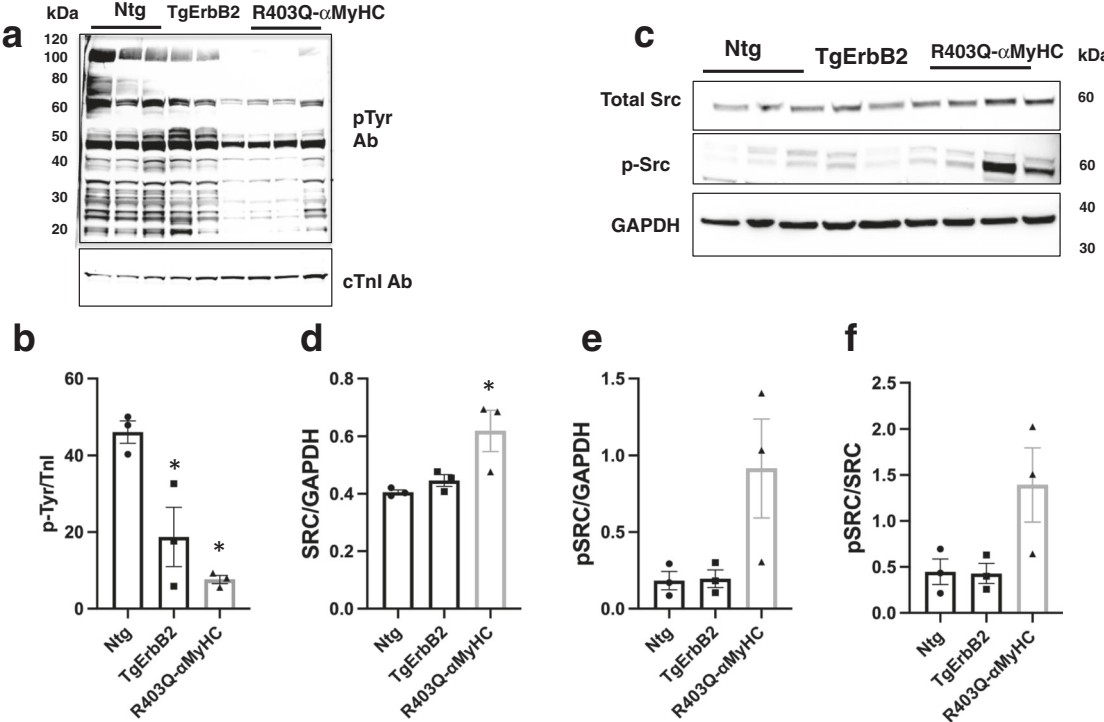

**Fig. 1 Global pTyr is reduced in cardiac hypertrophy associated with ErbB2 overexpression or myosin heavy chain mutation. a** Western Blots show a decrease in pTyr on TgErbB2 ($p = 0.0298$) mice and R403Q-α-MyHC ($p = 0.003$) heart homogenates. The Troponin I signal shows corresponding total protein loading. **b** Densitometry analysis shows that after the pTyrosine signal was normalized to total cTnI, the TgErbB2 and R403Q-α-MyHC mice significantly decreased in pTyr. **c** Western Blots show expression levels of total Src and p-Src (Tyr 416) and GAPDH as a loading control. **d** Densitometry analysis revealed that Src levels are significantly elevated in R403Q-α-MyHC ($p = 0.0282$), whereas **e** and **f** Show that p-Src phosphorylation levels are elevated in R403Q-α-MyHC but did not reach statistical significance. Signals in **d** and **e** were normalized to total GAPDH, whereas in **f** pSRC was normalized to total SRC. Ntg ($n = 3$), TgErbB2 ($n = 3$), and R403Q-α-MyHC ($n = 3$), error bars represent S.E.M. One-way ANOVA was used to compare groups.

undertook bioinformatic analyses to define the impact of ErbB2 cardiac overexpression and R403Q mutation of the cardiac myosin heavy chain on the specific pTyr changes. We determined the signaling transduction pathways associated with two unrelated types of HCM. Spectral signal normalization of pTyr was performed over the whole global heart TMT isobaric probes that quantified full proteome (Fig. 2e, f), and statistical methods were performed, as previously described[21,22]. Then, principal component analysis (PCA) and hierarchical clustering were applied to determine if the pTyr profiles were similar within the groups. This approach revealed that the three groups segregate by principal component 1 (PC1 = 46%) in full proteome quantification (Fig. 3a). This finding was also evident after pTyr normalization to full proteome; the three groups segregate along the principal component (PC1 = 52.2%) (Fig. 3b). Of note, one technical/biological replicate of R403Q-αMyHC was removed because it had more than 50% absent pTyr peptides compared to the other two replicates. Hence, it was considered a technical failure. Heatmaps of hierarchically clustered up- or down-regulated normalized pTyr peptides helped visualize the technical reproducibility and the specificity of how the genotype of cardiac hypertrophy largely influenced the pTyr proteome remodeling (Fig. 3c, $p < 0.05$ by ANOVA). Tyrosine-phosphorylated peptides are color-coded according to their extracted chromatogram MS1 signal intensities normalized to TMT isobaric label peptide MS2 intensity. Yellow is upregulated, and blue is downregulated. This approach provided us with the global heart proteome tyrosine peptide abundance. These results demonstrate that Ntg, TgErbB2, and R403Q-αMyHC mice have a pTyr proteome with a distinct signature,

evidenced by PCA and heatmaps clustering in unsupervised unbiased statistical methods.

Statistically significant changes (Log$_2$ fold change >1, -Log $p$-value < 1.3) were detected in 23 phosphosites corresponding to 18 proteins in TgErbB2 mice (Fig. 3d) and 45 phosphosites in 35 proteins in the R403Q-αMyHC Tg mice (Fig. 3f) using ANOVA ($p < 0.05$). It is noteworthy that pTyr of several peptides of the alpha-myosin heavy chain (*Myh7*-Y386, Y410, Y1375), one peptide of the beta-myosin heavy chain (*Myh6*-Y1349), and two peptides of Titin (*Ttn*-Y2118, Y21190) are downregulated while cardiac *Ttn* Y-31324 pTyr is upregulated on TgErbB2 mice hearts (Fig. 3d). Interestingly, *Myh7*-Y410 phosphorylation is downregulated in TgErbB2 and upregulated in R403Q-αMyHC Tg mice hearts (Fig. 3e). In contrast, cTnI Y113 (*Tnni3*-Y113) pTyr is downregulated in both HCM models but only reached statistical significance in R403Q-αMyHC Tg mice hearts. Titin phosphorylation (*Ttn*Y1881, Y1901, and Y33864) was significantly downregulated in the latter model.

We used MATLAB, a custom-made PhoshoEnrichment software (M. Ayati), which integrates the Molecular Signatures Database (MsigDB)[23] and a novel database of PTMs site-specific phosphorylation signatures of kinases, perturbations, and signaling pathways (PTMsigDB)[24]. The Broad Institute created the databases, which comprehend a well-curated collection of annotated gene data sets and PTMs. PTMsigDB accounts for phospho site-specific changes and their impact on activating or inhibiting a given pathway, together with the most common experimental systems biology perturbations. The advantage of these bioinformatics tools is to assess the statistical significance of each pathway and perturbation (gene-level and

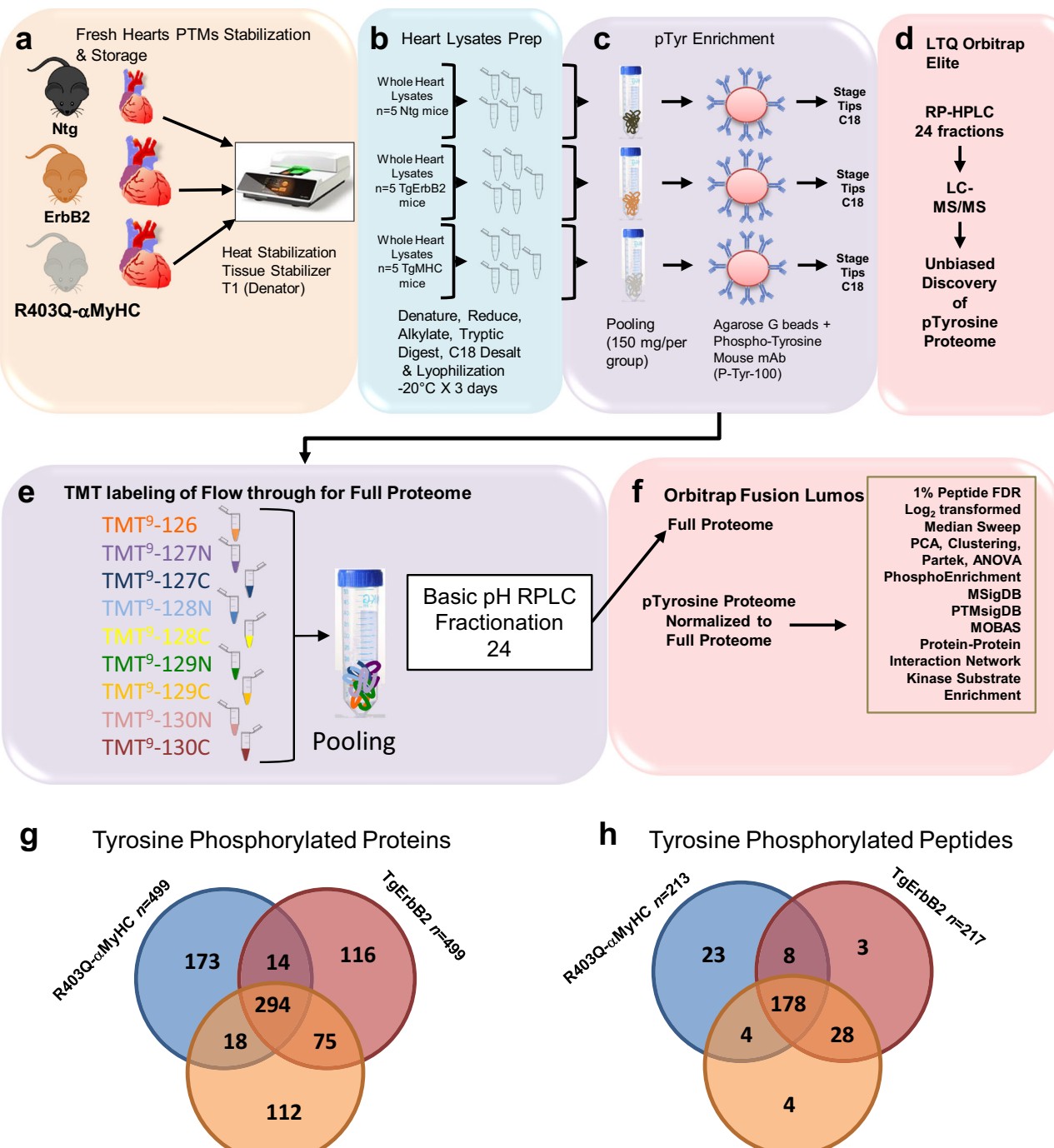

**Fig. 2 Workflow for global whole heart TMT phospho-tyrosine proteomics profiling. a** Ventricular tissues were rinsed with PBS and subjected to Heat Stabilization using a T1 Tissue Stabilizor to maximize phosphorylation preservation, then snap-frozen in liquid nitrogen and stored at −80 °C until further use. **b** Ntg, TgErbB2, and R403Q-αMyHC mice hearts ($n = 5$ per group) were processed in parallel to obtain whole heart lysates; 30 mg of protein lysates were used from each mouse heart for in-solution trypsin digestion, a total of 150 mg of trypsinized protein per genotype were C18 desalted and lyophilized at −20 °C for three days. **c** A total of 150 mg of tryptic peptides (pooled material from 5 hearts per group) were enriched for pTyrosine using immunoaffinity precipitation (Protein Agarose Beads + 300 micrograms of mAb Anti-pTyr) and desalted on C18 stage tips. **d** RP-HPLC ESI (electrospray ionization) MS/MS was performed on an LTQ Orbitrap Elite (Thermo Scientific) for a 120 min on a linear acetonitrile gradient (4−40%). Raw data were searched with Mascot 2.3, and label-free quantification with MS1 extracted ion chromatograms were performed using MaxQuant. This approach was repeated three times with samples from all three groups running parallel; 15 hearts per genotype in three technical replicates. **e** A fraction of the flow-through of pTyr enrichment was used for a 9-plexTMT; three replicates of each genotype (Ntg, TgErbB2, and R403Q-αMyHC) were pooled, fractionated in RPLC for full proteome quantification, **f** Data from pTyrosine and Full Proteome were processed in a similar way, pTyrosine signals were normalized to matched total protein expression to normalize data. Analyses are detailed in the text and supplemental data. **g** Shows the overlap of phosphorylated proteins found in the three groups, whereas **h** shows the overlapping tyrosine-phosphorylated peptides among the three groups.

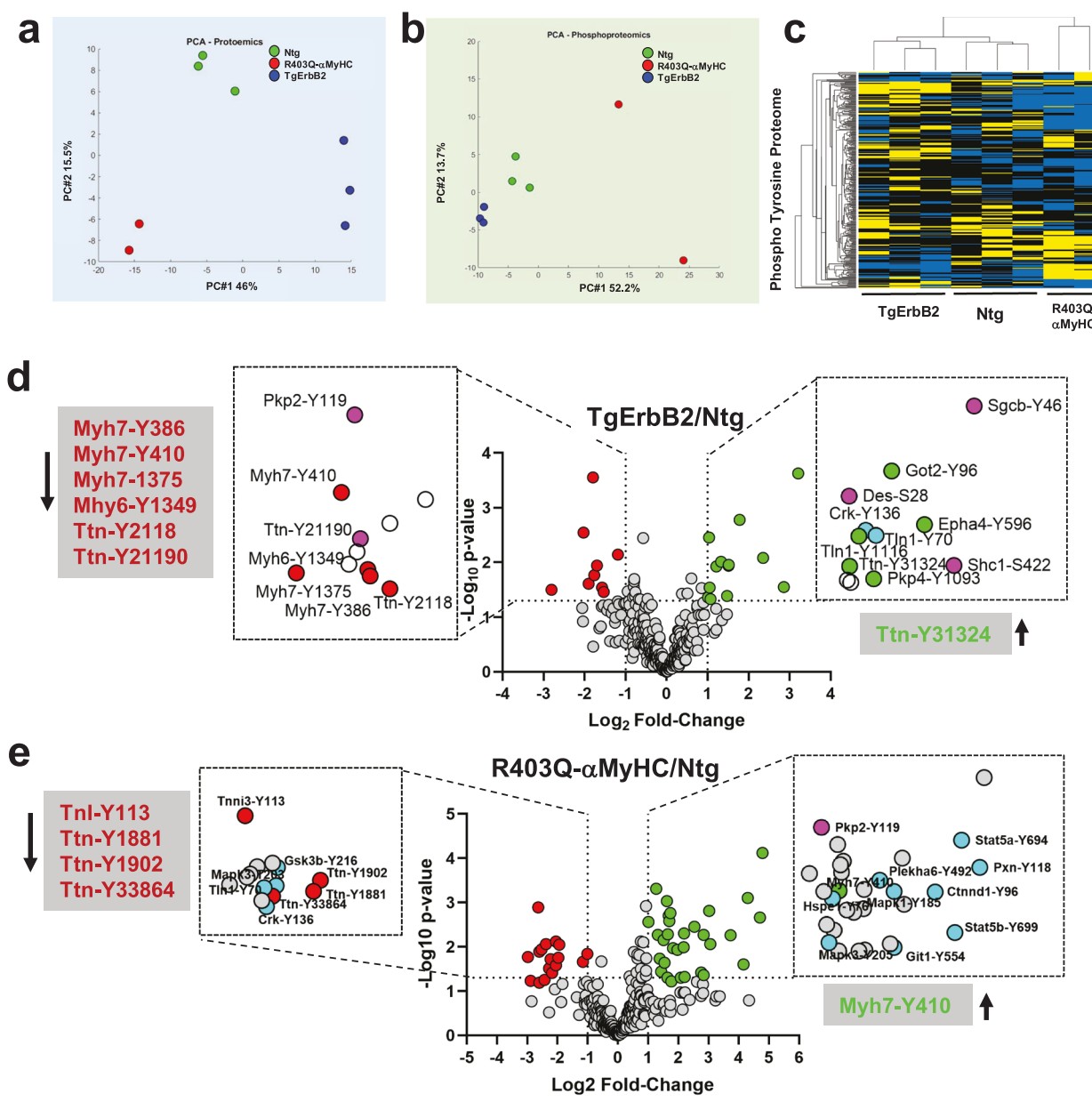

site level) using a hypergeometric test. Here, the number of identified phosphosites and genes is used as the population for the hypergeometric model parameter instead of all known genes.

A pathway enrichment analysis was performed on significant genes/sites using a gene-protein level with MSigDB. TgErbB2 mice's significant pathways are arrhythmogenic right ventricular cardiomyopathy (ARVC), HCM, DCM, and Integrin AlphaIIB Beta3 Signaling (Fig. 3f). R403Q-αMyHC Tg most significant pathways are the angiopoietin receptor pathway, ErbB signaling pathway, growth hormone receptor signaling, and chemokine signaling pathway (Fig. 3f).

**Fig. 3 PTMSigDB reveals common pTyr signatures of cardiac proteome in HCM. a** Unsupervised PCA of the full cardiac Proteome of Non-transgenic, TgErbB2, and R403Q-αMyHC mice show distinct segregation of the experimental groups. The groups segregate along with the first principal component (PC #1), which accounts for 46% of the total correlation in the expression of Full Proteome (normalized, mean $= 0$, variance $= 1$). **b** Unsupervised PCA of the pTyrosine peptides signal normalized to full proteome expression. The groups also segregate along with the first principal component (PC #1), accounting for 52.2% of the total correlation. **c** A heatmap was used to visualize the unsupervised hierarchical clustering of pTyrosine peptides normalized to full proteome expression with a $p$-value $< 0.05$ by ANOVA. Overrepresented (yellow) and underrepresented (Blue). **d** Volcano plot showing the Log2 fold-change and –Log10 p-value of each pTyr peptide in the pairwise comparison of Ntg and TgErbB2. The data points highlighted in red show a significantly lower intensity, whereas the data in green show significantly higher intensity. The significant sites were magnified in the figure (dotted squares) to label some peptide sites. Magenta data points match genes significantly associated with MsigDB disease pathways. Cyan data point did not reach statistical significance although point to EGFR1 pathway. **e** Volcano plot showing the Log2 fold-change and –Log10 p-value of each pTyr peptide in the pairwise comparison of Ntg and R403Q-αMyHC. Red and green data points were color-coded as above. With a similar magnification as above, cyan data points show a significant association to EGFR1 Pathway by PTMsigDB. **f** Molecular Signature DB (MSigDB) and Molecular PTMs Signature DB (PTMsigDB) of Global pTyr data identified pathways that were significantly altered.

Both Tg animal models showed a remarkable alteration in plakophilin-2 (*PKP2*) pTyr at the site Y119, although in TgErbB2 hearts, it was down-regulated, and in R403Q-αMyHC Tg hearts, it was upregulated (Fig. 3e). MsigDB analysis of global whole heart TMT flow through the entire proteome is displayed in Supplementary Data 3. *PKP2* is a critical component of the myocardium's desmosomes. Mutations in this gene are associated with arrhythmogenic cardiomyopathy[25–27]. Therefore, disturbed phosphorylation of Plakophilin-2 could have a functional impact.

In comparison, the PTMsigDB revealed that pathway analysis at the phosphorylated site-specific level did not reach statistical significance for the EGFR1 Pathway in TgErbB2 mice; only two site-specific phosphosites matched the EGFR1 pathway; they are highlighted in cyan color at the Volcano plot (Fig. 3d) and PTMsigDB table (Fig. 3f). Intriguingly, in R403Q-αMyHC mice, the same pathway was statistically significant; see the ten site-specific changes that matched the EGFR1 pathway ($p = 0.001$) represented in the volcano plot as cyan data points (Fig. 3e) and PTMsigDB table (Fig. 3f). These data show the utility of phosphorylation site-specific databases to narrow the search for a biologically relevant pathway on phosphoproteomics data sets, particularly in understudied phosphorylation events such as pTyr with small data sets to date.

**TMT-labeled quantitative proteomics of TgErbB2 confirms cardiac sarcomere tyrosine phosphoproteome dysregulation.** We used a TMT quantitative labeling proteomics to gain more specific insight into the myofilament pTyr changes. More specifically, we hypothesized that myofilament enrichment would enhance the number of site-identification in myofilament proteins, especially those with low abundance phospho-tyrosine modifications that may have been missed using the global approach. To this end, we utilized myofilaments freshly isolated from Ntg and TgErbB2 mice, as described in "Methods"[28], with minor modifications. We used three hearts per genotype to characterize the myofilament pTyr proteome (Fig. 4a, b), as described for the heart failure phosphoproteome[20]. Briefly, we performed a large-scale 6-plex TMT experiment where most of TMT-labeled peptides were enriched for pTyr (Fig. 4c). We adopted a workflow similar to whole heart TMT phospho-tyrosine proteomics for further statistical data analysis (Fig. 4d). The peptides identified with single spectra were removed, which led to 24,727 PSM. After median sweep normalization, 1116 peptides corresponded to 1,092 proteins. See Supplementary Fig. 4a, b for spectral intensity distribution before and after normalization (Supplementary Fig. 4c, d). For the pTyr proteome, 4391 PSM were collected and followed the same quality control curation of the entire proteome to remove missing data and unique spectra, which led to 3064 PSM. Raw data of pTyr proteome spectral intensity distribution are shown before

normalization (Supplementary Fig. 5a, b) and after normalization (Supplementary Fig. 5c, d) and PCA (Fig. 5b). After median sweep normalization, 832 peptides corresponded to 184 different proteins. Specific pTyr for 146 peptides was quantified because they had corresponding peptides from the full proteome's expression data.

The first step of the analysis uses unsupervised PCA and hierarchical clustering. PCA analysis showed the correlation between members in the same group (Fig. 5a–c), full proteome, pTyr proteome, and normalized pTyr proteome segregate well along PC1, showing a correlation of expression of 57.6%, 82.4%, and 62.1%, respectively. Heatmaps of hierarchically clustered expression helped visualize how TgErbB2 cardiac overexpression largely influenced the genotype clustering. Both patterns, proteins from the whole proteome (Fig. 5d) and the normalized intensity of pTyr proteome (Fig. 5e), suggested a mirrored remodeling (377 proteins for the entire proteome and 51 pTyr peptides had a $p < 0.05$ by LIMMA moderated 2-sample t-test comparison).

The full proteome detected 1116 peptides which corresponded to 1092 proteins. A comparison of the $Log_2$ fold-change (FC) of Ntg/TgErbB2 showed 377 proteins with statistically significant differences ($Log_2$ FC $> 1$ and $p$-value $< 0.05$). Like whole global heart TMT phospho-tyrosine normalized to full proteome data, the statistically significant protein expression changes were subjected to MSigDB for pathway analysis. The results are shown in Supplementary Data 4.

We used 146 phospho-sites corresponding to 50 proteins to analyze the pTyr proteome phosphorylation or normalized pTyr proteome. The TgErbB2 mice showed significant changes in 21 pTyr sites (Fig. 5g) corresponding to 15 proteins. A substantial increase of pTyr was detected on *MLC1*-Y82, Y139, *Myh6*-Y1310, α-*Tm*-Y261, *Actin*-Y168, *MyBP-C*-Y544, and *Actinin2*-Y200, among other proteins. The MsigDB pathway analysis yielded results similar to the global TMT phospho-tyrosine approach on whole hearts (Fig. 3). TgErbB2 sarcomere pTyr data also show that the most significant pathways are DCM, and HCM (Fig. 5g, h). Interestingly, when normalized to total ErbB2, the *ErbB2*-Y1006 phosphorylation levels decrease. On the other hand, the PTMsigDB pathways analysis did not match any pathways because many sites are new and not reported in the PTMsigDB database.

Our data show that the enrichment of sarcomere proteins detected more myofilament pTyr peptides that were missed in the global approach of whole heart lysates. However, the total number of tyrosine-phosphorylated peptides and corresponding proteins was lower.

**KSEA implicated downregulation of MAPK in TgErbB2 and upregulation of EGFR in R403Q-αMyHCTg.** KSEA was used to characterize genotype-induced signaling changes by estimating

## Strategy for Cardiac Sarcomere Tyrosine Phosphoproteome Profiling of Normal and ErbB2Tg Myofilaments by TMT Quantitative Proteomics

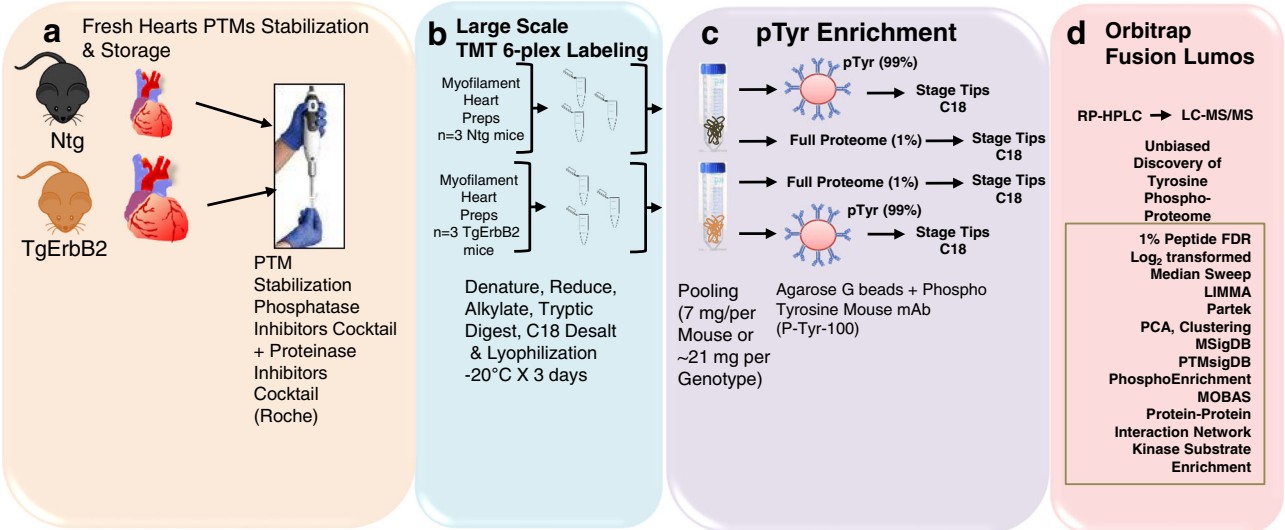

**Fig. 4 Workflow for cardiac sarcomere pTyr identification of normal and TgErbB2 hypertrophic myofilaments by TMT quantitative proteomics.**
**a** Myofilament from Ntg ($n = 3$) and ErbB2 ($n = 3$) transgenic mice hearts were freshly isolated on ice-cold buffers containing Proteinase and Phosphatase Inhibitors Cocktails (Roche). **b** All material was resuspended in TEAB, then reduced and alkylated. Tryptic peptides were desalted and labeled with 6-plex isobaric tandem mass tags (TMT). **c** The digested and labeled peptides were pooled and desalted with $C_{18}$ SEP-PAK. The enrichment for phosphotyrosine was performed with PTMScan Phospho-Tyrosine Rabbit mAb (P-Tyr-1000) kit (Cell Signaling Technology). The eluted peptide samples were desalted using C18 STAGE tips **d** Easy-nanoLC 1200 nanoflow liquid chromatography system coupled to Orbitrap Fusion Lumos Tribrid.

the kinase's relative activity in TgErbB2 and R403Q-αMyHC mice using the global whole heart pTyr TMT and TMT quantitative sarcomere pTyrosine proteomics data, using their respective Ntg groups as a reference. KSEA[29] is a method that infers the kinases' differential activity based on the differential phosphorylation of its substrates and computes scores that reflect the directional change in each kinase's activity. This method assumes that the differential activity of kinases is correlated with phosphorylation changes in its substrates. A positive score corresponds to a kinase with phosphorylated substrates in Tg mice relative to Ntg control. Likewise, a negative score is a hypophosphorylation in Tg relative to their Ntg control. The kinase-substrate interaction data was downloaded from the PhosphoSitePlus[30] website. Next, the KSEA method was applied to all the pTyr sites identified in these experiments. We identified 34 phosphosites that have associated kinases reported on PhosphoSitePlus. However, 49 unique kinases were scored in the combined data sets. A KSEA heatmap inferred clusters of kinases downregulated (blue color scale) or upregulated (red color scale) is shown in Fig. 6a. TgErbB2 mice displayed significant downregulation of *MAP2K4*, *MAP3K6*, *MAP3K5*, *MAP2K3*, and *MAP2K6*, and marked upregulation of *EPHA4*. On the other hand, R403Q-αMyHC Tg showcased a substantial downregulation of *GSK3B* (see highlighted blue rectangle in Fig. 6a). R403Q-αMyHC Tg displayed a larger cluster of significantly upregulated kinases (see highlighted red rectangle in Fig. 6a).

Conventional pathway analysis can miss protein groups because pathway algorithms are predefined and rigid. A protein–protein interaction (PPI) network approach might better capture signaling responses in these models that are not typically detected by pathway analysis. To do so, MoBaS Analysis[22,31] was employed to identify densely connected subnetworks that are related and might exhibit differential phosphorylation in Tg models. Several modules were identified, and we focused on the top two statistically significant PPI modules from the global whole heart TMT flow through dataset; for interpretation

purposes, they were designated as Module 1 and Module 2. The substrates of modules 1 and 2 are illustrated in Fig. 6b, d, respectively.

Interestingly, PTMsigDB site-level molecular signature analysis showed that module 1 contained significant site-specific modifications for the EGFR1 pathway. In addition, using MSigDB Gene-Protein Level analysis, several pathways, such as the ErbB signaling, PDGFRA, and focal adhesion pathways, showed statistical significance. KSEA of module 1 protein-protein network highlights significant activation of *AURKB* (aurora kinase B) and *CSNK2A1* (Fig. 6c).

Similarly, PTMSigDB analysis for module 2 confirmed the involvement of EGFR1 Pathway while MsigDB detected actin cytoskeleton, ErbB signaling, and focal adhesion pathways. KSEA of module 2 protein-protein network highlights significant activation of *SRC*, *MAP2K1*, *HCK*, *LCK*, *SYK*, *JAK3*, *JAK2*, *FYN*, *PTK6*, and *ABL1* in R403Q-αMyHC mice (Fig. 6e). These data confirm that MoBAS identified functionally relevant modules of PPI among identified pTyr peptides, and PTMsigDB pointed to EFGR1 Pathway as a common pathway for HCM from two unrelated mouse models.

**Tyrphostin AG-825 administration decreased cardiomyocyte disarray in the TgErbB2 mouse model.** TgErbB2 mouse hearts develop a concentric type of HCM. This alteration rapidly progresses to pathologic HCM, showing features such as fibrosis, cardiomyocyte disarray, perturbed $Ca^{2+}$ handling, arrhythmias, and sudden death[16]. On these grounds, we hypothesized that counteracting the activity of cardiac ErbB2 overexpression pharmacologically, a tyrosine kinase receptor, using AG-825 (an ErbB2 inhibitor), would halt the progression of the histopathological phenotype. To that end, TgErbB2 mice (ages 6–9 months) with established HCM and Ntg controls were treated with AG-825 or DMSO vehicle for two weeks, subcutaneously twice daily at a dose of 1 mg/Kg[16]. Next, myocardial fibrosis was evaluated by

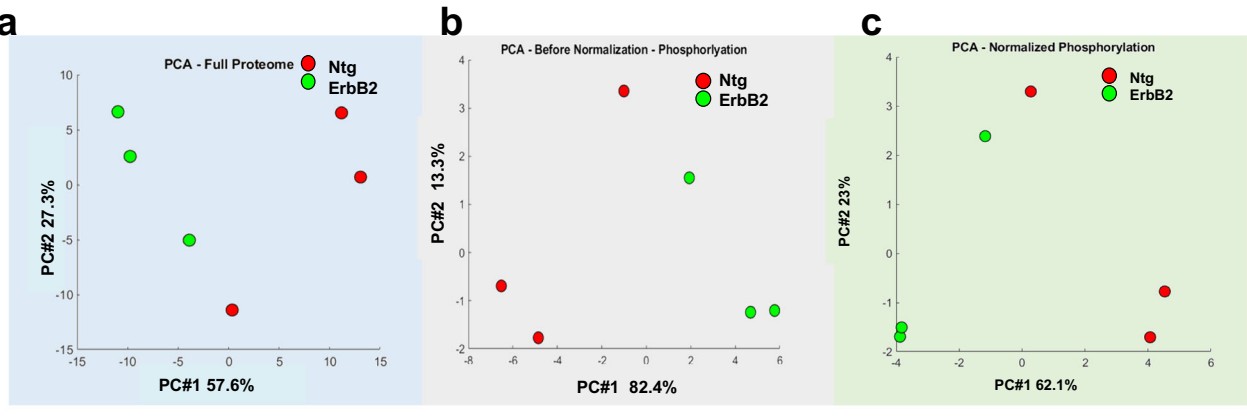

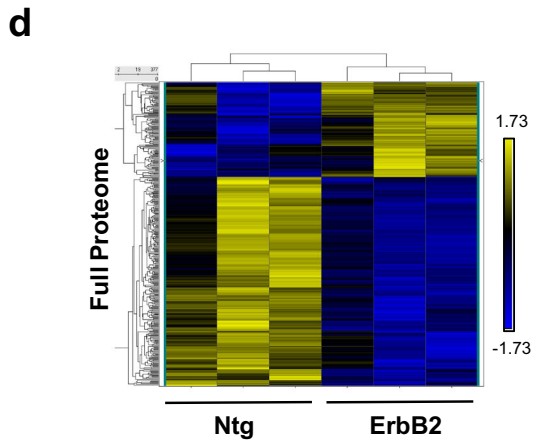

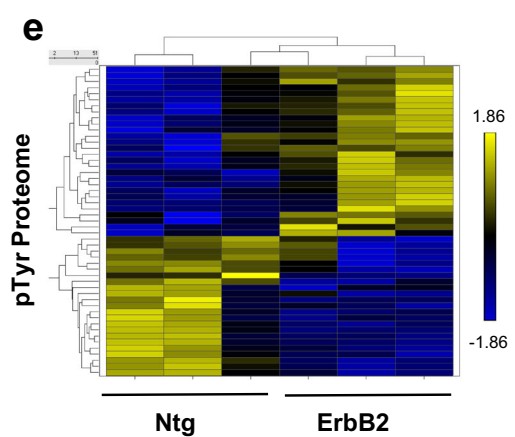

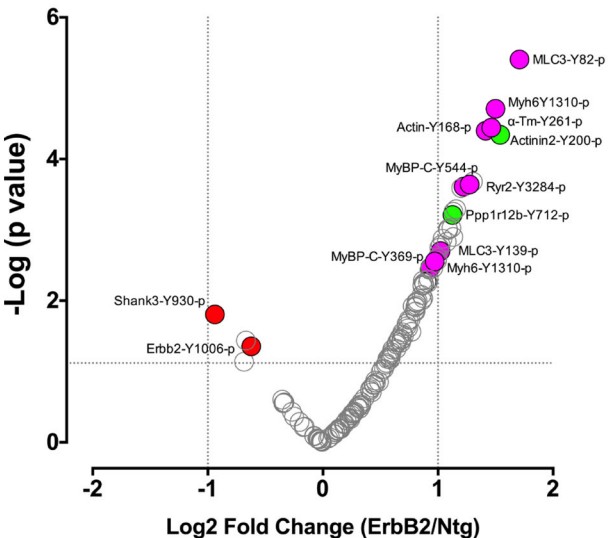

**f**  **Ntg/TgErbB2 Normalized pTyr Proteome**

**g**  **Molecular Signature (MSigDB) Gene-Protein Level**

| Pathway KEEG | Genes in pathway | p-value |
|---|---|---|
| DILATED CARDIOMYOPATHY | 'MYH6','TPM1','MYL3','MYBPC3','ACTC1','RYR 2','TTN' | 3.3E-3 |
| HYPERTROPHIC CARDIOMYOPATHY HCM' | 'MYH6','TPM1','MYL3','MYBPC3','ACTC1','RYR 2','TTN' | 3.3E-3 |

Masson's trichrome staining. Untreated TgErbB2 mice developed variable amounts of fibrosis of the left ventricle (Fig. 7a, b); however, TgErbB2 treated mice displayed less fibrosis (Fig. 7b). In contrast, cardiac histology was not affected by AG-825 Treatment in Ntg mice. We used CytoSpectre[32], a web graphic user interphase, to characterize the effects of AG-825 on cardiomyocyte disarray; this interphase determines the local orientation of structures, including cardiac myocytes, across histological

sections, similar to the analysis described previously by the Seidman's laboratory[33]. Randomly chosen regions of interest are depicted in Fig. 7a, and increase the magnification of microphotographs from 5X to 40X (Fig. 7b). The myocyte orientations angle variance was evaluated, referred to as Circular Variance and means orientation angles variance. Ntg treated with the vehicle (Fig. 7c) compared to Ntg AG-825 treated (Fig. 7c) was not significantly different (Fig. 7d), as already described[33]. In

**Fig. 5 TgErbB2 cardiac sarcomere pTyr proteome remodeling. a** Full Proteome was labeled with TMT, and the logged and summarized protein signal intensities indicate that experimental groups segregate along with the first principal component (PC #1), which accounts for 53.9% of the total correlation in the expression proteome (normalized, mean $= 0$, variance $= 1$). **b** PCA of the phospho-Tyr Proteome reveals a similar trend, and the groups segregate along the PC#1, which accounts for 47.2% of the correlation. **c** Signal intensities of the phospho-Tyr proteome were normalized to Full Proteome protein expression and analyzed by principal component analysis. The distinctive signature of pTyr is still apparent by PC#1 and accounts for 57% of the correlation. **d** A heatmap was used to visualize the unsupervised hierarchical correlation clustering of protein expression and **e** pTyr Proteome expression with $p < 0.05$ by LIMMA moderated 2-samples t-test comparison of TgErbB2 and Ntg group. **f** Volcano plot showing the Log2 fold-change and –Log10 p-value of each pTyr peptide in the pairwise comparison of Ntg and TgErbB2. The data points in red showed a significantly lower intensity, whereas the data points in green showed a significantly higher intensity. For clarity of presentation, only some peptides were highlighted. **g** Molecular Signature DataBase (MSigDB) confirmed targets and the involvement of Dilated and HCM KEEG Pathways also found on TgErbB2 mice in Fig. 3.

contrast, TgErbB2 mice cardiomyocytes treated with the vehicle displayed an orientation angle variance that was substantially different from Ntg-Vehicle control; see the heterogeneous and broader shape of the orientation plot in Fig. 7c. Surprisingly, cardiomyocytes from TgErbB2 mice AG-825 treated displayed significantly less disarray, as demonstrated by the significant return of circular variance and shape of the plot to Ntg- Vehicle or AG-825 treated controls (Fig. 7e, $p = 0.0014$ by One-way ANOVA). This analysis suggests that cardiomyocytes from TgErbB2 mice AG-825 treated displayed significantly less myocyte disarray, as indicated by reduced circular variance.

Next, we evaluated the direct in vivo impact of ErbB2 inhibition by AG-825 on contractility, echocardiograms, and tissue-doppler imaging to assess diastolic function. These studies were performed on conscious (non-anesthetized) mice, at baseline and after two weeks of Treatment. We used a Vevo 2100 High-Resolution Imaging System with a 40 MHz transducer (VisualSonics®, Toronto). The data were subsequently analyzed with an Advanced Cardiovascular Package Software. The parameters studied were chamber dimensions, fractional shortening, ejection fraction, and tissue Doppler velocity dynamics. Left ventricle functional data obtained by parasternal short-axis echocardiographic imaging determined no significant change in the mean fractional shortening over time in TgErbB2 mice treated with vehicle solution compared with the Ntg (Fig. 7f). Similarly, TgErbB2 mice treated with AG-825 did not display significant changes in contractile function over time compared to Ntg AG-825 treated (Fig. 7f). Tissue Doppler imaging detected no significant differences in diastolic function; see complete Echocardiography data in Supplementary Data 6. These data suggest that ErbB2 pharmacological inhibition halted or reversed pathological remodeling by reducing the fibrotic response and restoring cardiomyocyte alignment. Yet, this response was not paralleled by in vivo preservation of LV contractile function in TgErbB2 mice.

**Extensive phosphoproteomics validate EGFR1 pathway activation on cardiac hypertrophy of TgErbB2 that is inhibited by tyrosine kinase inhibitor (AG-825).** We revisited the phosphoproteomics of TgErbB2 and NTg to validate bioinformatic findings of KSEA and to study the impact of the tyrosine kinase inhibitor AG-825 on the proteome. We used a TMT 13-plex labeling approach to compare the whole heart phosphorylation profile of Ntg and TgErbB2 mice treated with vehicle or AG-825 (ErbB2 inhibitor). Briefly, a TMT 13-plex was performed as follows; Ntg-Vehicle ($n = 3$), Ntg-AG-825 treatment ($n = 3$), TgErbB2-Vehicle ($n = 3$) and TgErbB2-AG-825 treatment ($n = 4$). Whole heart protein lysates were isolated from thermally stabilized hearts and processed as described in Fig. 2 and "Material and methods". Most of the material was used to enrich pS/pT/pY peptides, and a small portion is labeled to quantify full proteome with TMT. As described in the previous experiment in Fig. 2, the intensity of pS/pT/pY peptides is normalized to the

intensity of full proteome TMT signals. A total of 6614 proteins, 4732 phosphorylated serine and threonine peptides, and 346 pTyr peptides were quantified by TMT reporter ions. The full dataset is available in Supplementary Data 7.

Heatmaps of hierarchically clustered up- or down-regulated normalized serine/threonine phosphorylated, or tyrosine-phosphorylated peptides intensity confirm the influence of the genotype on all phosphoproteome remodeling, serine and threonine phosphorylation (Fig. 8a) and pTyr (Fig. 8b). We directly interrogated data with MSigDB and PTMSigDB to compare Ntg with TgErbB2 mice. As anticipated, we confirmed the involvement of EGFR1 Pathway by PTMSigDB in TgErbB2 mice cardiac hypertrophy treated with vehicle (Fig. 8c, d) or AG-825 (Fig. 8e, f). Also, we confirmed the involvement of focal adhesion and extracellular matrix pathways by MSigDB in vehicle-treated mice and dilated cardiomyopathy, HCM, and arrhythmogenic right ventricle cardiomyopathy (ARCV) in AG-825 treated mice. However, AG-825 Treatment for 14 days had a subtle effect in TgErbB2 mice that was not evident by MsigDB or PTMsigDB. Since we included all phosphoproteome, we estimated kinase activity in serine and threonine phosphorylation peptide. The top upregulated and downregulated kinases were identified using KSEA, and the result is presented in Fig. 8g. The complete KSEA and pathway analysis are available in Supplementary Data 5 and Supplementary Data 7. We compared the activity of kinases in AG-825 treated animals relative to their control (vehicle treatment). We identified a small cluster of kinases significantly inhibited or downregulated by AG-825 Treatment and not by vehicle treatment (See highlighted red rectangle in Fig. 8g). To get more insights on the functional impact from a system biology point of view, we explored the relation of these kinases using The Signaling Network Open Resource (Signor 2.0) visualization software[34]. Figure 8h shows that most of these kinases are downstream of EGFR, highlighted in blue dotted circles. Altogether, our data indicate that EGFR1 is central to hypertrophy of TgErbB2 and that AG-825 Treatment inhibits ErbB2 and EGFR; therefore, a significant subset of targets downstream of EGFR signaling cascade are affected.

## Discussion

The present study demonstrates that the pTyr proteome of the heart is altered in proteins related to cardiomyopathies, serine/threonine, and tyrosine kinases in two pathogenetically-unrelated models of hypertrophy cardiomyopathy (HCM), namely cardiac-specific overexpression of ErbB2 and allelic expression of R403Q-αMyHC. In addition, the serine/threonine phosphoproteome of the heart is also altered in the TgErbB2 mouse model cardiac hypertrophy. Altogether our data point that EGFR pathway activation is centrally involved in cardiac hypertrophy that rises from two very different mouse models.

Heart failure contractile dysfunction is, in part, due to myofilament $Ca^{2+}$ desensitization. Altered serine/threonine

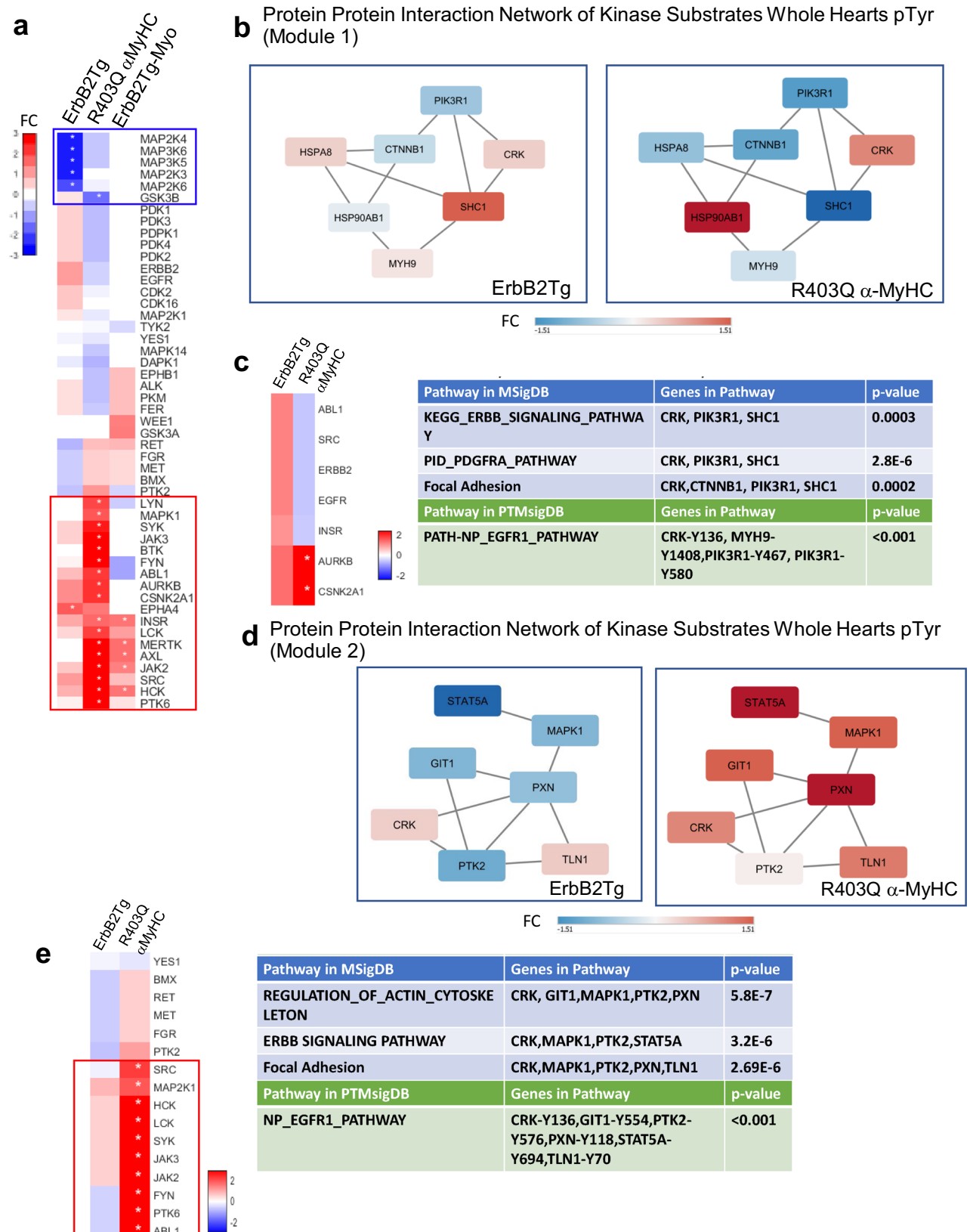

**b** Protein Protein Interaction Network of Kinase Substrates Whole Hearts pTyr (Module 1)

**d** Protein Protein Interaction Network of Kinase Substrates Whole Hearts pTyr (Module 2)

| Pathway in MSigDB | Genes in Pathway | p-value |
|---|---|---|
| KEGG_ERBB_SIGNALING_PATHWAY | CRK, PIK3R1, SHC1 | 0.0003 |
| PID_PDGFRA_PATHWAY | CRK, PIK3R1, SHC1 | 2.8E-6 |
| Focal Adhesion | CRK,CTNNB1, PIK3R1, SHC1 | 0.0002 |
| Pathway in PTMsigDB | Genes in Pathway | p-value |
| PATH-NP_EGFR1_PATHWAY | CRK-Y136, MYH9-Y1408,PIK3R1-Y467, PIK3R1-Y580 | <0.001 |

| Pathway in MSigDB | Genes in Pathway | p-value |
|---|---|---|
| REGULATION_OF_ACTIN_CYTOSKELETON | CRK, GIT1,MAPK1,PTK2,PXN | 5.8E-7 |
| ERBB SIGNALING PATHWAY | CRK,MAPK1,PTK2,STAT5A | 3.2E-6 |
| Focal Adhesion | CRK,MAPK1,PTK2,PXN,TLN1 | 2.69E-6 |
| Pathway in PTMsigDB | Genes in Pathway | p-value |
| NP_EGFR1_PATHWAY | CRK-Y136,GIT1-Y554,PTK2-Y576,PXN-Y118,STAT5A-Y694,TLN1-Y70 | <0.001 |

phosphorylation of the cardiac sarcomere and reduced myofilament $Ca^{2+}$ sensitivity are features of human and experimental models of heart failure and play a role in the pathophysiology of HCM and DCM[7,35,36]. We have previously identified pTyr in human cardiac Troponin I Y26 that was downregulated in human heart failure and DCM[14]. However, no studies have yet addressed

the impact of global dysregulated cardiac pTyr in the pathogenesis of cardiac disease.

Here, we show evidence that in TgErbB2 mice's heart, tyrosine phosphoproteome is altered in a differential fashion, upregulation, and downregulation of tyrosine phosphopeptides. MsigDB pathway analysis demonstrated the involvement of ARVC, HCM,

**Fig. 6 KSEA & MoBaS analysis. a** KSEA on whole hearts pTyr enrichment and TMT pTyr Myofilament enrichment data. Heat maps report KSEA results according to normalized scores (i.e., TgErbB2/Ntg, R403Q-αMyHC/Ntg, etc.). Only kinases shared between data sets are included, therefore present in Tg and Ntg. Asterisks indicate the statistical significance of $p < 0.05$. Red represents a positive score, and blue is a negative score. Red rectangle highlights groups of activated kinases clustered together and are predominant in R403Q-αMyHC, whereas the blue rectangle highlights suppressed kinases that clustered and are predominant on TgErbB2 whole hearts. **b** MoBaS identified a tightly interconnected subnetwork involved in growth factors signaling (ERBB, PDGFRA, and EGFR1). Represented by the top-scoring PPI subnetwork called Module 1 was identified on TgErbB2 whole heart pTyr enrichment and compared to R403Q-αMyHC. **c** KSEA on Module 1 group of substrates, MsigDB and PTMsigDB. **d** Second top-scoring PPI subnetwork called Module 2 was identified on R403Q-αMyHC whole heart pTyr enrichment and compared to TgErbB2. **e** KSEA on Module 2 group of substrates show activated kinases in R403Q-αMyHC. Results from PTMsigDB and MSigDB analysis identified EGFR1, Actin cytoskeleton, ERBB and Focal Adhesion signaling. Each node is a phosphoprotein represented by its most significant peptide on PPI modules. Node color is based on Log2 fold-change relative to NTG. The scale is noted by FC and a bar colored from blue to red.

DCM, and integrin alphaIIB beta3 signaling in TgErbB2 mice hearts. Most of the sarcomere proteins had significant down-regulation of pTyr, such as *Myh7*-Y386, Y410, Y1375, Myh6-Y1349, and Ttn-Y2118, Y21190, except for Ttn-Y31324 that was upregulated. The directional change of pTyr in these sarcomere proteins is downregulation, just like it was reported for human cTnI Y26 in heart failure and DCM. Very little is known about the role of pTyr of sarcomere proteins. In fact, only one group[37] has examined the potential tyrosine kinases that phosphorylate cTnI. They demonstrated in vitro that c-Src and another Src kinases family member, Lyn, phosphorylates the cTnIY26, reporting phosphorylation and pseudo-phosphorylation mimicking at this site reduced myofilament Ca$^{2+}$ sensitivity, as evidenced by the altered force-calcium response of the myofilaments. Noteworthy, cTnI Y113 phosphorylation, a novel dis-covered site, was downregulated in R403Q-αMyHC mice. We also demonstrated significant downregulation of pTyr in *Ttn*-Y1881, Y1902, and Y33864. Contrary to TgErbB2 mice, *Myh7*-Y410 was upregulated in R403Q-αMyHC mice. pTyr at *Myh7*-410 is near R403Q αMyHC and could have a functional impact since it is in the motor domain head of αMyHC. The site-specific implications of these changes are worthy of further study; however, it is beyond the scope of the present work. Interestingly, in the R403Q-αMyHC hearts, pTyr proteome is also altered differently. However, in R403Q-αMyHC mice, more phosphotyrosine pep-tides reached statistical significance. Unlike TgErbB2, the MsigDB pathway analysis demonstrated that R403Q-αMyHC mice had alterations in tyrosine kinase receptor pathways, angiopoietin receptor, ErbB, and growth hormone receptor signaling. Whereas PTMsigDB revealed the alteration of the EGFR1 Pathway and squamous cell carcinoma. We reasoned that overexpression of a receptor tyrosine kinase in TgErbB2 mice hearts would result in a more significant alteration of pTyr proteome. However, the data suggest that a single sarcomere point mutation, as in the case of R403Q-αMyHC mice, is incisive enough to alter the cardiac pTyr proteome significantly. Top-down proteomics[38] demonstrated that consistent alterations of cTnI, ENH2, and other Z-disk protein phosphorylation in HCM myocardium correlated better with the phenotype, regardless of the single sarcomere point disease-causing mutation. The mechanisms that a pathogenic mutation in the myosin heavy chain (R403Q-αMyHC) may affect pTyr regulation are unknown and warrant further investigation.

Global pTyr dysregulation in the TgErbB2 mouse model is essential because it connects pronounced cardiac hypertrophy, sarcomere dysfunction, and abnormal calcium handling[16] with alterations in tyrosine kinase pathways. On the other hand, pharmacological inhibition of ErbB2 (HER2/neu) with the inhi-bitor lapatinib in breast cancer patients treated with doxorubicin increases the risk of developing heart failure compared to patients treated with doxorubicin alone[39]. It suggests that maintaining pTyr homeostasis may be necessary for regulating cardiomyocyte function and homeostasis.

A wide variety of other functionally relevant targets were identified in this study with pTyr, from sarcomere proteins to z-disk, intermediate filaments (such as desmin), desmosome com-ponents, focal adhesion, and adherence junction proteins, mem-brane receptors, kinases, and phosphatases. For instance, pTyr in several z-disk-associated proteins was noted (for a complete list, see Supplementary Data 2, 3, and 6). Z-disk proteins are crucial for muscle contraction and mechanical stress, growth, and metabolic signaling[40]. Also, the alteration in the phosphorylation levels of the desmosome key component Plakophilin-2 protein (*Pkp2*) peptide *Pkp2*-Y119 was noted in both mouse models, upregulation in R403Q-αMyHC Tg and downregulation in TgErbB2. The functional effect of up- or downregulation of *Pkp2*-Y119 phosphorylation is not known. However, *Pkp2* homozygous deletion disrupts heart architecture and is lethal in the embryo[41]. In the heart, *Pkp2* is required to assemble the desmosome and the PKC activity[42]. Autosomal dominant mutations in this gene are responsible for 25 to 50% of ARVC. Interestingly, TgErbB2 mice have increased susceptibility to arrhythmias and myofibrillar disarray[16], similar to patients with myosin mutations and HCM. *Pkp*-Y119 phosphorylation changes could impact the phenotype.

The EGFR1 (ErbB1) pathway plays a central role in cardiac hypertrophy in both Tg models. EGFR pharmacological inhibi-tion, using AG-1478, protects against Angiotensin II-induced cardiac hypertrophy in vitro and in vivo[43]. The concentric hypertrophy associated with ErbB2 cardiac-specific over-expression can be reversed by early administration of Lapatinib, which inhibits EGFR receptor tyrosine kinase[12] in addition to ErbB2. In this study, the ErbB2 receptor's pharmacological blockage by AG-825 in TgErbB2 adult mice reduced myocyte disarray but did not preserve cardiac function or alter cardiac hypertrophy. We think that the lack of response is related to the age of mice at the time of administration: in the current study, we used adult mice, whereas pups have been used in previous studies[15]. We also evaluated the cardiac serine/threonine and pTyr proteome in TgErbB2 after vehicle or AG-825 Treatment. We found global alterations in the phosphoproteome (Fig. 8) that confirmed the involvement of the EGFR pathway. In addition, we confirmed the alterations in focal adhesion, extracellular matrix, DCM, HCM, and ARCV pathways. We used KSEA to pair the findings with a subset of activated kinases and narrow it down to the top 40 most representative serine/threonine and tyrosine kinases detected in the cardiac proteome of Ntg and TgErbB2 mice. AG-825 Treatment could inhibit or significantly reduce the activation of 14 kinases corresponding to the EGFR pathway (Fig. 8g, h), suggesting that AG-825 Treatment was effective and specific. Altogether these findings indicate that alterations of cardiac proteome span to serine/threonine and pTyr, confirming the involvement of the above-described pathways. In addition, these data suggest that AG-825 treatment-induced changes in the phosphoproteome correlated with an improvement in cardiac myocyte disarray.

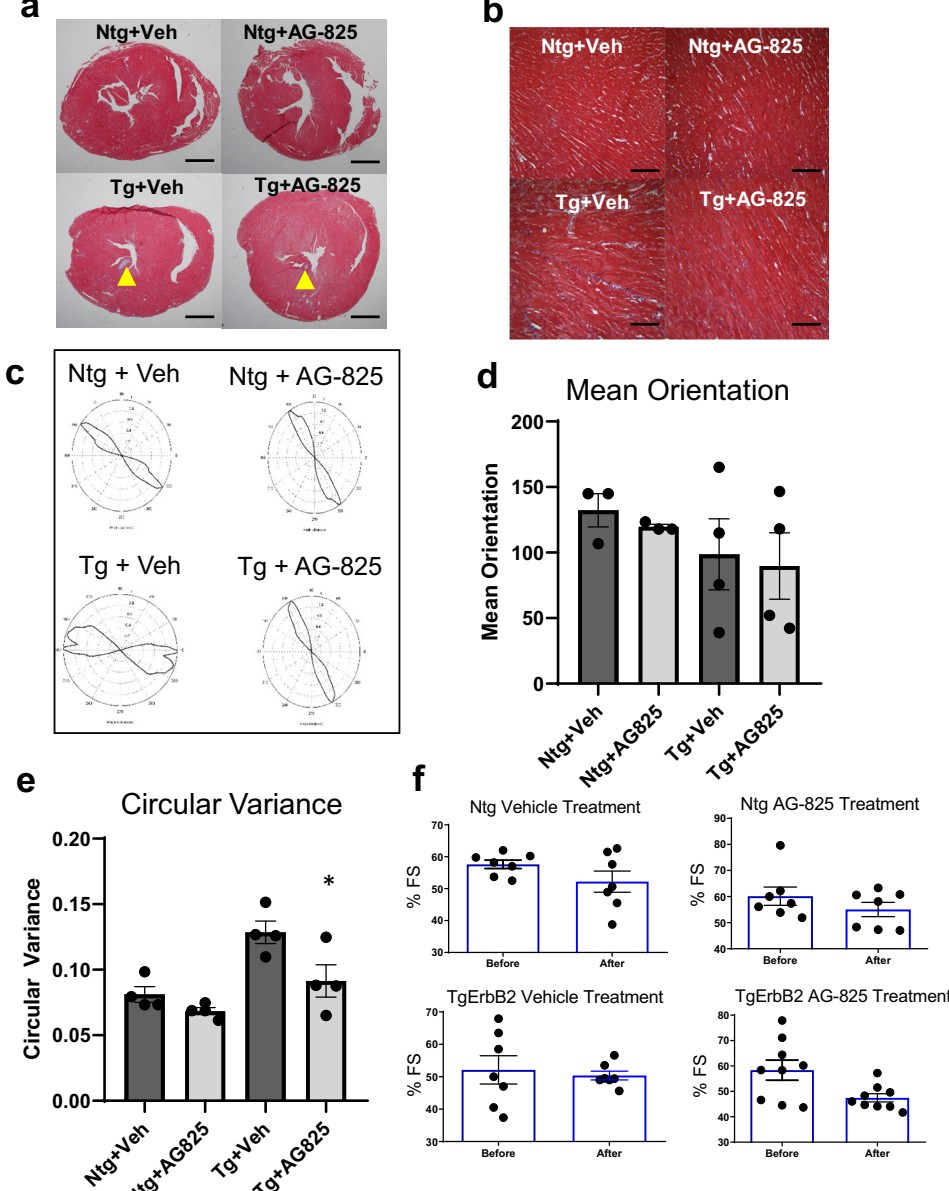

**Fig. 7 Tyrosine kinase receptor inhibitor AG-825 reduces cardiac myocyte disarray without effects on contractile function in TgErbB2 mice.**
**a** Masson's trichrome stained sections (5X) from vehicle-treated Ntg and AG-825–treated Ntg and vehicle-treated TgErbB2 and AG-825-treated TgErbB2 mice, yellow arrowheads point to fibrotic areas. Scale bar 1 mm. **b** Masson's trichrome stained sections (40X) from representative regions of interest show increased Fibrosis in TgErbB2 mice myocardium, that fibrosis improved after 2 weeks of AG-825 Treatment. The same areas were used to determine the level of cardiac myocyte disarray, which was achieved by analyzing cell orientation using CytoSpectre Software from Vehicle and AG-825 treated Ntg and TgErbB2 mouse heart sections (scale bar 50 µm). **c** Plots of the distribution of cardiac cells orientation angles. Note how Ntg angles are homogeneous and did not change in the group treated with AG-825. In contrast, Tg treated with vehicle showed less homogenous and different orientation angles, confirming cardiomyocyte disarray. AG-825 Treatment reduces cardiomyocyte disarray, and the orientation plot is similar to the AG-825 treated NTG group. **d** Mean orientation angle of cardiac myocytes, and **e** Circular variance, another measure of isotropy, in Ntg and TgErbB2 mice with and without AG-825 treatment ($n = 5$ animals per condition). Graph bars are expressed as mean ± s.e.m.; a one-way ANOVA test was performed for statistical comparisons showing a significant improvement in TgErbB2 mice treated with AG-825 (*p*-value < 0.0014). f Cardiac echocardiography from the four groups shows that AG-825 Treatment for 14 days did not modify the contractile function, estimated as % Fraction Shortening.

The KSEA implicated a marked downregulation of members of the canonical MAPK pathway downstream of *k-Ras* in TgErbB2 mice. Conversely, in R403Q-αMyHC mice, we observed an upregulation of focal adhesion and PDGFR-beta signaling pathways (*SRC, LYN, LCK, JAK2, INSR, MAPK1*, and *HCK*). More importantly, modularity-base scoring (MoBas) analysis identified protein-protein interaction subnetworks enriched in ErbB, PDGFRA, and focal adhesion pathways and activation of aurora kinase B and *CSNKA1*. These data indicate that many targets overlap in both transgenic models of cardiac hypertrophy. Although there are marked differences in the upstream regulators of ErbB2 and myosin heavy chain, many downstream effector

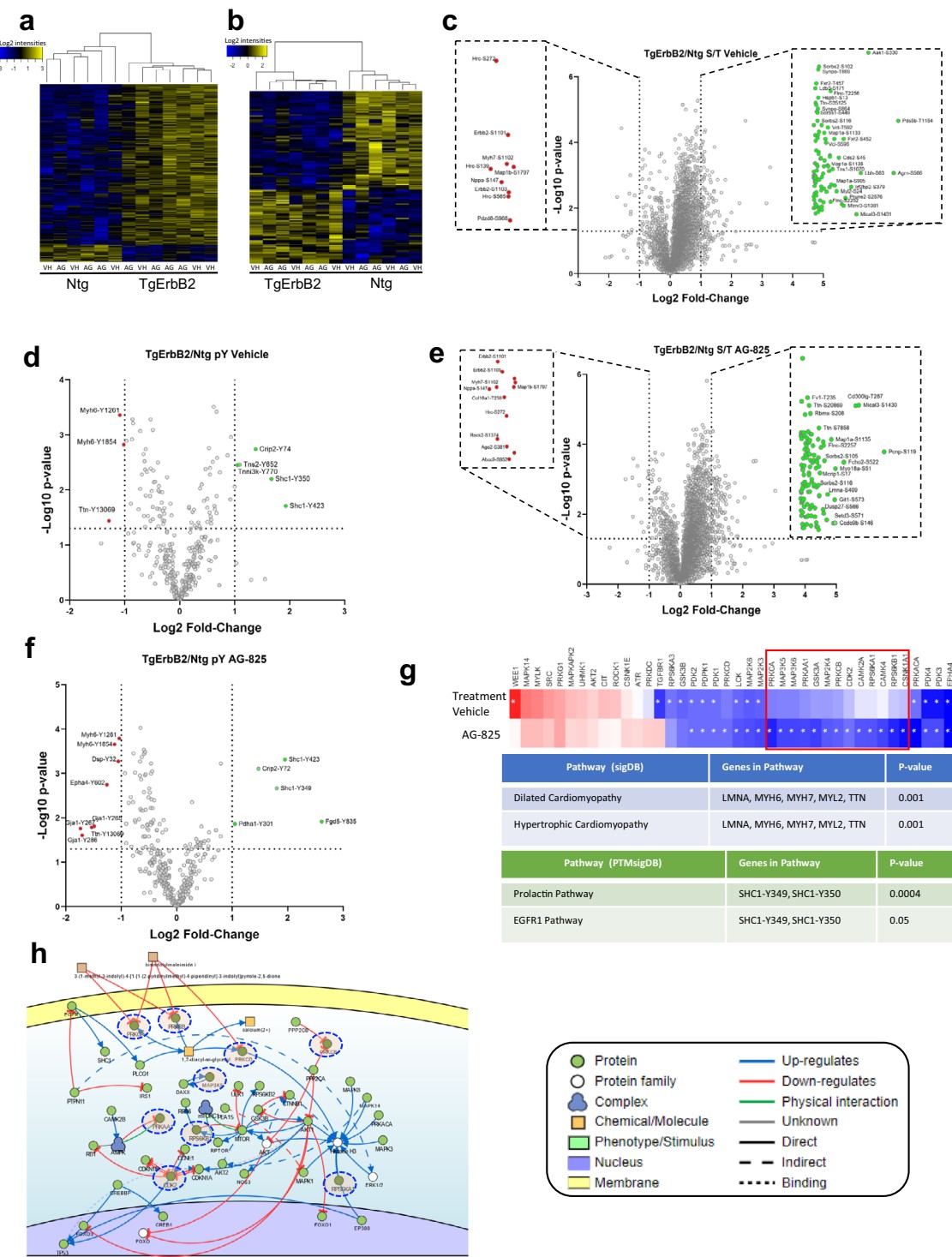

**Fig. 8 Comprehensive serine/threonine/tyrosine phosphoproteomics confirm EGFR1 pathway involvement in TgErbB2 hypertrophy and tyrosine kinase receptor inhibition by AG-825. a** A heatmap was used to visualize the unsupervised hierarchical clustering of Ser/Thr peptides, and **b** pTyrosine peptides normalized to full proteome expression with *p*-value < 0.05 by ANOVA. Overrepresented (yellow) and underrepresented (Blue). **c** Volcano plots showing the Log2 fold-change and –Log10 *p*-value of each pSer/pThr peptide and **d** pTyr peptides in the pairwise comparison of Ntg and TgErbB2 after two weeks of Vehicle Treatment. **e** Volcano plots showing the Log2 fold-change and –Log10 *p*-value of each pSer/pThr peptide and **f** pTyr peptides in the pairwise comparison of Ntg and TgErbB2 after two weeks of AG-825 Treatment, (**g**) KSEA shows in a red rectangle the kinases significantly inhibited by AG-825. KSEA and Pathway Enrichment Analysis of combined Ser/Thr/Tyr phosphorylation data analysis using PTMSigDB confirm that EGFR1 Pathway is significantly involved. **h** The Signaling Network Open Resource (Signor 2.0) analysis shows that nine kinases are involved in the EGFR pathway. Kinases found in the network and also in KSEA are circled in dotted blue circles. The symbol code of the network is displayed next to the pathway.

molecules are common and can be targeted by small-molecule inhibitors (designed initially to treat several types of cancer) to block cardiac hypertrophy.

The MAPK pathway is involved in the adaptive and maladaptive response that could lead to heart hypertrophy (for review[44]). The RAS-RAF-MEK-ERK signaling pathway is an attractive target for oncology therapeutic intervention. Several selective *RAF* and *MEK* small-molecule inhibitors have been tested in clinical trials[45]. The present study found increased phosphorylation at *MAPK1*-Y185, which activates the kinase in R403Q-αMyHC, and *MAPK1* activity directly relates to heart hypertrophy[46]. The c-Src, the central kinase from its family, could be one of the potential regulators for the pTyr proteome changes observed in R403Q-αMyHC because it phosphorylates the EGFR receptor upstream and phosphorylates Stat, among other targets, downstream. c-Src affects the response to mechanical cardiomyocyte stretching by triggering a cascade of intracellular signaling in cardiomyocytes towards a hypertrophic response[47,48]. Also, c-Src phosphorylates *PXN*-Y118[49], and this site had a 25-fold increase in phosphorylation in R403Q-αMyHC mice. Interestingly, c-Src mediates the activation of *MAPK1* and *MAPK3* in response to pressure-overload[47]. Notably, a previous study has shown that pressure-overload-induced cardiac hypertrophy is exacerbated in R403-αMyHC Tg mice[50], suggesting that the mechanism by which R403Q-αMyHC mutation produces heart hypertrophy is by sensitizing cells to pressure-overload-induced signaling via c-Src.

R403Q-αMyHC mice displayed enhanced *JAK2, STAT5A,* and *STAT5B* phosphorylation in the activation sites (Y570, Y694, and Y699, respectively), which indicates not only activation of MAPK signaling but also activation of JAK-STAT signaling. Jak-Stat Signaling: IL-6 pathway is activated in response to the IL-6 family of cytokines (IL-6, cardiotrophin 1, and leukemia inhibitor factor), cross-talks with the EGFR pathway, and is involved in cardiac hypertrophy. This pathway has cardioprotective effects, but chronic activation may lead to heart hypertrophy (reviewed in ref. 44). Vakrou and colleagues performed a pathway analysis on miRNA profiles of R403Q-αMyHC. These authors found similarities with the findings described in this work, such as overactivation of chemokine signaling (CXCR4), actin cytoskeleton, and cardiac hypertrophy signaling pathways[51]. These results suggest an essential involvement of pTyr regulation in myofilaments and other cardiac proteins. The insight gained from these studies could inform new therapeutic approaches in sarcomere mutation-related HCM and potentially other conditions associated with HCM. The studies were limited to two transgenic mouse models that develop cardiac hypertrophy; other models of sarcomere mutations or pressure-overload such as aortic banding could be explored and compared. Some relevant tyrosine-phosphorylated sites might be missed due to their low stoichiometry and technical challenges. This problem is particularly true for membrane-bound proteins, which are difficult to evaluate in phosphoproteomic studies.

This study reveals that altered pTyr patterns are striking in two separate models of HCM. Moreover, despite some shared sites, etiologically different forms of HCM may harbor specific pTyr signatures that point EGFR pathway as central in both transgenic models. This evidence, combined with the inhibition of a specific receptor tyrosine kinase using tyrphostin AG-825, can reverse cardiomyocyte disarray and rationalize approaches to manipulate the pTyr proteome as a therapeutic approach for HCM. Furthermore, these studies indicate that tyrosine kinase inhibitors, now used broadly in cancer therapies, may change cardiac function by directly modifying the heart's tyrosine kinase and serine/threonine kinase profiles.

## Methods

**Western blot**. Whole heart lysates from Heat Stabilized Tissue were resuspended in 1% SDS buffer, 10–15 μg were separated by SDS-PAGE, and transferred to nitrocellulose membranes. Membranes were blocked with 5% BSA in TBS-T buffer (20 mmol/L Tris pH 7.4, 150 mmol/L NaCl, and 0.1% Tween 20) for 1 h at room temperature, then were incubated with primary antibody dilution 1:1000 in 1% BSA-TBS-T; phospho-Tyrosine mouse mAb (pTyr-100, cat. No. 9411SCST), TnI Rabbit Ab (cat. No. 4002SCTS), Src Rabbit Ab (cat. No.2108SCST), Phopsho-Src Family (Tyr416) Rabbit mAb (D49G4, cat. No.6943T CST), GAPDH Rabbit Ab (14C10, cat. No. 2118S CST) at 4 °C overnight. After washing them five times, the secondary antibodies were diluted 1:10,000 in 1% BSA-TBS-T (anti-Rabbit IgG-HRP linked (cat. No. 7074S CTS), Anti-Mouse m-IgGk BP-HRP (cat. No. sc-516102, Santa Cruz) and incubated for 1 h at room temperature. Then, the membranes were developed with super signal West Pico Chemiluminescent Substrate (Thermo), and the immunoreactive bands were detected by chemiluminescence with an iBright 1500 (Invitrogen). The images were obtained and analyzed with Image J software. One-way ANOVA was used to compare groups and determine statistical significance.

### Global whole heart TMT phospho-tyrosine proteomics

*Sample preparation*. All protocols were performed following the "Guide for the Use and Care of Laboratory Animals" published by the National Institutes of Health and the Institutional Animal Care and Use Committee's approval. The TgErbB2 (B6SJLF1/J stain) and R403Q-αMyHC (C57/Bl6) mice were obtained from Dr. K. Gabrielson and Dr. L. A. Leinwand, respectively[16,17], to establish breeding colonies. Male or female mice (6-9 months) were anesthetized with sodium pentobarbital IP (75 mg/kg) or isoflurane (5%) overdose; the hearts were quickly dissected, followed by thermal stabilization (Denator T1 Heat Stabilizor, Sweden) and stored at −80 °C until analyzed. TgErbB2 and R403Q-αMyHC mice hearts ($n = 5$ per group) were processed in parallel. To obtain whole heart lysates, cardiac ventricles (~200 mg) were homogenized on ice-cold buffer: 20 mM HEPES, pH 7.6, 1 mM, 1.5 mM sodium pyrophosphate, PhosStop, and 9 M urea at 10 μl/mg (wet weight of tissue). The homogenate was cooled on ice, followed by brief micro-tip sonication on ice, centrifugation at 10,000 × *g* for 15 min at 4 °C. The supernatant was retrieved, and protein concentration was determined by the method of Lowry (Bio-Rad).

*Trypsin digestion*. Protein from heart lysates was reduced in 5 mM of dithiothreitol (DTT) at 60 °C for 20 min and alkylated in 10 mM iodoacetamide (IDA) at room temperature for 15 min in the dark. Each sample (30 mg per mouse, $n = 5$ mice per genotype) was digested with Proteomics grade Trypsin (Promega) at a ratio of 1:200 in 2 M urea, 20 mM HEPES buffer, pH 8.0 at room temperature overnight. The digestion was terminated with trifluoroacetic acid (1% TFA). Samples were centrifuged (5 min at 1800g), and supernatants were desalted by solid-phase extraction (SepPak C18 10cc cartridge, Waters). Elutes were lyophilized for three days at −20 °C.

*Enrichment of tyrosine phosphopeptides*. A total of 150 mg of trypsinized peptides per genotype were pooled and enriched for pTyrosine, as previously described[52]. Lyophilized peptides were mixed in 1.4 ml immunoprecipitation buffer (IAP buffer 50 mM MOPS, pH 7.2, 10 mM sodium phosphate, 50 mM NaCl, pH 7.0). A stock solution of protein agarose G beads (Santa Cruz Biotechnology) 80 μL of slurry were conjugated with 300 μg of Phospho-Tyrosine Mouse monoclonal antibody (p-Tyr-100, Cell Signaling Technology). The beads-Anti-p-Tyr antibody conjugate was transferred to the peptide's tube and incubated with gentle rotation for 2 h at 4 °C. The beads were washed and eluted with 55 μl and 45 μl of 0.15% tri-fluoroacetic acid (TFA), respectively. The two elution yields were pooled. The resulting peptide mixtures were purified by solid-phase extraction (stage tips C18, Thermo Scientific). The samples were dried by vacuum centrifugation. This approach was repeated three times with samples from all three groups run in parallel, 15 hearts per genotype in three technical replicates. The immunoprecipitation buffer flow-through was stored at −80 °C for subsequent TMT 9-plex. According to the manufacturer's instructions, tryptic peptides were desalted and labeled with 9-plex isobaric TMT (Thermo Scientific). The labeling reaction was carried out for 1 h at room temperature, followed by quenching with 100 mM Tris.HCl (pH 8.0).

*LC-MS/MS analysis*. Phosphopeptides were dissolved in 10 μl of 0.1% TFA, 2% ACN (v/v) followed by RF-HPLC-ESI-MS/MS analysis. Phosphopeptides were separated on a C18 reversed-phase column with a linear gradient of acetonitrile (4−40%) for 120 min and then analyzed on an LTQ-Orbitrap Elite MS (ThermoFisher Scientific) with neutro loss triggered HCD.

*Peptide identification and quantification*. Raw MS data were searched with Mascot 2.3, and label-free quantification with MS1 extracted ion chromatograms were performed using MaxQuant software.

### TMT from flow-through for total proteomics

*LC-MS/MS analysis and database search*. Peptides were analyzed on Orbitrap Fusion Lumos Tribrid (Thermo Scientific) coupled with Easy-nanoLC 1200

nanoflow liquid chromatography system (Thermo Scientific). The peptides from each fraction were reconstituted in 10% formic acid and loaded on Acclaim Pep-Map100 Nano Trap Column (100 μm × 2 cm) (Thermo Scientific) packed with 5 μm $C_{18}$ particles at a flow rate of 5 μl per minute. Peptides were resolved at 250 nl/min flow rate using a linear gradient of 10 to 35% solvent B (0.1% formic acid in 95% acetonitrile) over 120 min on the EASY-Spray column (50 cm × 75 μm ID, PepMap RSLC $C_{18}$, and 2 μm $C_{18}$ particles) (Thermo Scientific). It was fitted on an EASY-Spray ion source that was operated at 2.0 kV voltage. Mass spectrometry analysis was carried out in a data-dependent manner with full scans of m/z 350 to 1500. Both MS and MS/MS were acquired and measured using Orbitrap mass analyzer. Full MS scans were measured at a resolution of 120,000 at m/z 200. Precursor ions were fragmented using a higher-energy collisional dissociation method (35) and detected at a mass resolution of 30,000 at m/z 200. Proteome Discoverer 2.4 was used for identification and quantification against uni-port_mouse_120119_UP000000589. The search parameters used are as follows: (a) trypsin (up to two missed cleavages), (b) minimum amino acid length: 6, (c) minimum peptides for protein: 1, (d) Fixed modifications TMT6plex on any N-terminus and Lysis residue, Carbamidomethylation of cysteine, (e) Dynamic modifications: Oxydation on Methionine, acetyl on N-terminus, Met-loss on M and Met loss +acetyl on M and (f) 1% false discovery rate for peptide and protein levels.

## TMT cardiac sarcomere phospho tyrosine proteomics

*Sample preparation and TMT pTyr enrichment.* Myofilament enriched preparations were obtained by rinsing freshly isolated hearts with ice-cold PBS (Proteinase Inhibitors Roche 1X and PhosStop 1X, Roche), auricular tissue removed and ventricular tissue minced in ice-cold standard buffer (60 mM KCl, 30 mM Imidazole pH 7.0, 2 mM $MgCl_2$, Proteinase Inhibitors Roche 1X and PhosStop 1X from Roche). The ventricular tissues were homogenized with a homogenizer (Omni tissue homogenizer TH) at maximum speed for 2–3 pulses of 10 seconds with the solution on ice, centrifuged at 12,000 g for 15 min. The myofilament pellets were resuspended in a skinning solution (10 mK EGTA 8.2 mM $MgCl_2$, 14.4 mM KCl, 60 mM imidazole pH 7.0, 5.5 mM ATP, 12 mM creatine phosphate, 10 U/mL creatine phosphokinase, 1% Triton 100X, Proteinase Inhibitors Roche 1X and PhosStop 1X from Roche) and incubated on ice for 30 min[28]. Myofilament pellets were centrifuged at 1100 g for 15 min and washed in the standard buffer. Freshly isolated myofilament pellets were diluted in standard buffer and quantified with Lowery assay. Myofilament proteins (~7 mg) from each mouse heart were resuspended in 8 M urea and 50 mM triethylammonium bicarbonate (TEAB) (Sigma) followed by reduction with 10 mM dithiothreitol (Sigma) at room temperature for 1 h and alkylation with 30 mM iodoacetamide (Sigma) for 20 min in the dark, see Fig. 4a–d. The protein samples were then digested overnight at 37 °C using sequencing grade trypsin (1:50) (Promega). Tryptic peptides were desalted and labeled with 6-plex isobaric tandem mass tags (TMT) (Thermo Scientific) according to the manufacturer's instructions. The labeling reaction was carried out for 1 h at room temperature, followed by quenching with 100 mM Tris.HCl (pH 8.0). The digested and labeled peptides were pooled and desalted with $C_{18}$ SEP-PAK (Waters), followed by pTyrsoine enrichment using PTMScan® Phospho-Tyrosine Rabbit mAb (P-Tyr-1000) kit (Cell Signaling Technology), see Fig. 4c. Briefly, the desalted peptides were reconstituted in 1.4 ml of immunoaffinity purification (IP) buffer containing 50 mM MOPS pH 7.2, 10 mM sodium phosphate, 50 mM NaCl. Anti-phosphotyrosine antibody (p-Tyr-1000, Cell Signaling Technology) was mixed with peptide solution and incubated on a rotator at 4 °C for 2 h. After incubation, the beads were washed with IP buffer and water two and three times, respectively. The phosphotyrosine peptides were eluted using 0.1% TFA. The eluted peptide samples were desalted using C18 STAGE tips, vacuum dried, and kept at −80 °C before LC-MS analysis. From flow-through, a quantification of the full proteome was made to serve as a reference for pTyr proteome findings.

*LC-MS/MS analysis.* The phosphotyrosine peptides and non-phosphorylated peptides were analyzed on Orbitrap Fusion Lumos Tribrid (Thermo Scientific, San Jose, CA, USA) coupled with Easy-nanoLC 1200 nanoflow liquid chromatography system (Thermo Scientific). The peptides from each fraction were reconstituted in 10% formic acid and loaded on Acclaim PepMap100 Nano Trap Column (100 μm × 2 cm) (Thermo Scientific) packed with 5 μm $C_{18}$ particles at a flow rate of 5 μl per minute. Peptides were resolved at 250 nl/min flow rate using a linear gradient of 10–35% solvent B (0.1% formic acid in 95% acetonitrile) over 95 min on the EASY-Spray column (50 cm × 75 μm ID, PepMap RSLC $C_{18}$, and 2 μm $C_{18}$ particles) (Thermo Scientific) and it was fitted on EASY-Spray ion source that was operated at 2.0 kV voltage. Mass spectrometry analysis was carried out in a data-dependent manner with full scans in the range of m/z 350 to 1500. Both MS and MS/MS were acquired and measured using Orbitrap mass analyzer. Full MS scans were measured at a resolution of 120,000 at m/z 200. Precursor ions were fragmented using a higher-energy collisional dissociation method and detected at a mass resolution of 30,000 at m/z 200.

*Peptide search and Identification.* MaxQuant 1.5 was used for quantitation and identification against a mouse RefSeq database (version 78) supplemented with frequently observed contaminants. The search parameters used are as follows: (a)

trypsin as a proteolytic enzyme (with up to two missed cleavages); (b) peptide mass error tolerance of 10 ppm; (c) fragment mass error tolerance of 0.02 Da; (d) Carbamidomethylation of cysteine (+57.02 Da) and TMT tags (+229.16 Da) on lysine residue and peptide N-termini as fixed modification and oxidation of methionine (+15.99 Da) and phosphorylation of serine/threonine/tyrosine (+79.96 Da) as variable modifications. Two missed cleavages were allowed, and 'match between runs' was enabled. Peptides and proteins were filtered at a 1% false discovery rate. Proteins with a q-value lower than 0.05 will be considered as differentially expressed with statistical significance.

*Echocardiogram.* Cardiac morphology and function were assessed by transthoracic echocardiography using a high-resolution high-frequency system (Vevo 2100, VisualSonics, Canada) equipped with 40 MHz ultrasound probe in conscious mice[22,53]. Mechanical and chemical chest hair removal was performed by shaving and using a commercially available depilatory agent. Core temperature, monitored via a rectal probe, was maintained at 36.5–37 °C using a heating lamp. Parasternal long and short-axis views of the heart were captured in both B and M-modes at optimized frame rates. LV end-diastolic (LVIDd, mm) and end-systolic (LVIDs, mm) internal diameters, the end-diastolic wall thickness of both interventricular septum (IVS, mm) and LV posterior wall (LVPW, mm), LV ejection fractions (EF, %) and fractional shortening (FS, %) were measured using M-Mode in long-axis view. Relative wall thickness (RWT), a measure of hypertrophy, was calculated as IVSd+LVPWd/LVID. Pulsed-Doppler recordings at the mitral inflow, left ventricular outflow and basal septum was recorded. Digital images were stored and analyzed offline using commercially available Vevo LAB software (VisualSonics, Canada). An experienced echocardiography technician blinded to experimental groups performed all the measurements.

## Statistical analysis

*Western Blot densitometry.* The density of the bands was measured using ImageJ software and the area under the curve corresponding to the bands of pSRC, SRC, GAPDH, TnI, and total pTyr was calculated. Then, pTyr/TnI, SRC/GAPDH, pSRC/GAPDH, and pSRC/SRC ratios were calculated and the ratios were compared between groups using one-way ANOVA.

*Mean orientation and circular variance from histological samples comparison.* To characterize the effects of AG-825 on cardiomyocyte disarray, we used the web graphic user interphase CytoSpectre[32] to determine the local orientation of cardiomyocytes across histological sections. In randomly chosen regions, the myocyte mean orientation angle circular variance was calculated. Statistical comparison between groups (Ntg+Vehicle, Ntg+AG825, TgErbB2+Veh, TgErbB2 + AG825) was made with one-way ANOVA.

**Systems biology analysis.** Statistical analysis of the label-free dataset was performed using Perseus software[54]. Peptides with less than 50% of detection were filtered, and the missing values of the intensities of the reminding peptides were filled with k-nearest neighbors. Then, the log2 intensities were normalized by median subtraction, ANOVA, and false discovery rate calculations. A heatmap was used to visualize the unsupervised hierarchical clustering of proteins with a p-value <0.05 by ANOVA test. For statistical analysis of the TMT-labeled proteomics data collected, we followed the workflow of Foster et al.[21], with few modifications. First, Partek software was used to calculate each peptide intensity's p-values, q-values, fold change, and ratio. Phosphosites with less than 50% of missing density values were subjected to statistical analysis, and the reminding missing values were filled with the k-nearest neighbors. Then, a normalization of the Log2 densities by median subtraction and determined the fold-change, ratio, and statistical significance (q and p values), from each of the selected phosphosites of the dataset. The volcano plots were made with GraphPad Prism 7®. Unsupervised PCA was used in both data sets separately with Partek® software. The dataset was segregated along with each component concerning the pTyr abundance (normalized, mean = 0, variance = 1).

Normalization to the full proteome performed by log2 intensity subtraction was done, followed by a two-sided t test and the moderated p-values and q-values calculation using an algorithm developed by Herbrich et al.[55]; R software was used for this analysis.

*Pathway enrichment analysis.* The hypergeometric p-value was used to identify the significantly enriched processes in the proteins and phosphosites that are differentially phosphorylated between case and control samples. For this purpose, MsigDB[23] was used to analyze the protein level pathways and PTMsigDB[24] to analyze the processes at the site level. The population/background for which the enrichment is calculated is restricted to all the proteins in which their sites are quantified in the phosphoproteomics experiment, rather than all universally known proteins/phosphosites.

*KSEA.* The KSEA seeks to identify kinases whose targets exhibit significantly altered phosphorylation levels in a given condition. KSEA scores each kinase k with

a set of substrates S as follows:

$$score(k) = \frac{(\bar{P}_S - \bar{P}) * \sqrt{|S|}}{\sigma}$$

where $\bar{P}_S$ Den otes the average $\log_2$ of the fold change of all the substrates of kinase k, and $\bar{P}$ and $\sigma$ represent the average and standard deviation of $\log_2$ of the fold change of all the identified phosphosites in the dataset. KSEA was performed on the identified modules by restricting S to the substrates in the module instead of all substrates in the dataset. The data provided by PhosphoSitePLUS[30] was used as the reference for kinase-substrate associations. This tool is available from Dr. M. Ayati website: https://faculty.utrgv.edu/marzieh.ayati/software.html

*Module Identification.* First, networks were created in which nodes represent proteins and edges represent the PPI obtained from BioGRID[22,56]. The proteins were assigned a score in the networks by computing the average fold change of phosphosites residing on each protein obtained from experiments individually (i.e., Ntg, TgErbB2, and R403Q-αMyHC Tg). Then MoBaS[22] was applied to identify subnetworks of highly connected and differentially phosphorylated proteins. For visualization of subnetworks, the proteins are colored based on the average fold change of that protein in different conditions. If a protein in one dataset is not identified in another dataset, the node is represented in gray color.

*Echocardiography comparison.* One-way ANOVA was used to compare differences between groups to analyze the echocardiography data. For the effect of AG-825 or vehicle treatments, the pre- post- Treatment, we used a t-test.

**Statistics and reproducibility**. Experiments were conducted in replicates (see "Methods" section). Mascot (version2.3) was used for peptide identification from MS raw data, and MaxQuant (version 1.5) was used for quantification. Statistical analysis was performed using R (version 4.0), Partek Genomics Suit (version 7.0), GraphPad Prism 7.0, and Perseus (version 1.5). Additional details of data processing are described in the "Methods" section.

**Reporting summary**. Further information on research design is available in the Nature Research Reporting Summary linked to this article.

## Data availability

The mass spectrometry proteomics data have been deposited to the ProteomeXchange Consortium via the PRIDE[57] partner repository with the dataset identifier PXD036506. Supplemental Data 1 contains the numerical source of graphs presented in Figs. 1 and 7. Supplemental Fig. 6 contains the uncropped western blots of Fig. 1.

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

## Acknowledgements

The project described was supported by Grant Number K01-HL13368-01 to G.A.R.C., 5 T32 HL 7227-43, T32HL125239-05, T32HD044355 from NIH, R01 HL63038, R01 HL114910-01 and AHA 15GRNT25720026 to A.M.M., R01 HL13691 to N.P., R01 HL088649 to K.G. and R01LM012980 to M.A. Its contents are solely the authors' responsibility and do not necessarily represent the official views of the N.I.H. We thank Dr. Leslie Leinwand for facilitating the Transgenic Mice R403Q-αMyHCTg. National Science Foundation of China (81870364) and Research Foundation for the Scientific Development of Longgang district of Shenzhen (LGKCYLW2021000020), and Natural Science Foundation of Guangdong Province of China (2022A1515012468) supported M.X.

## Author contributions

M.X., A.M., H.B.K., X.Y., X.F., A.H., X.Z., and C.H.N. performed experiments and data analysis, A.E., K.G., D.B.F., and N.P. provided statistical support and critical review, K.C.B., M.A. provided bioinformatics and statistics support, M.X., K.C.B., N.P., A.M.M., and G.A.R.C. wrote the manuscript, A.M.M., and G.A.R.C. design the study and supervised the work.

## Competing interests

The authors declare no competing interests.
