## [Peer Review File · Communications Biology]

Reviewers' comments:

Reviewer #1 (Remarks to the Author):

The manuscript "Altered Tyrosine Phosphorylation of Cardiac Proteins Prompts Contractile Dysfunction in Hypertrophic Cardiomyopathy" investigates changes in the proteome and phospho-proteome of both the normal heart and the heart in two mouse models of heart disease that are believed to be related to phosphorylation. They are particularly focused on the role that tyrosine phosphorylation plays in the pathogenesis of these mutant strains. They find that each mutant affects tyrosine phosphorylation differently, with different predicted pathogenic effects. Using statistical modeling programs that are freely available, they were able to find enrichment or depletion of these phospho-tyrosines on many pathways and proteins involved in heart disease. Using this data set to guide them, they then treated mice in one of these heart disease models with a tyrosine kinase inhibitor and demonstrated that the inhibitor relieves the symptoms associated with this disorder. The work discussed is interesting and novel and presents a solid discovery from omics levels analyses of these heart disease models. However, I have a few questions that should be address, in general relating to the methodological approach of the authors.

- 1) When sampling the data, what was the time of day at which the mice were sac'ed? Was it at the same time for all the mice and if so, what was that time? The proteome and phospho-proteome of mice are known to change over the day and so differences in the time of sampling can have a significant effect on the proteins that are phosphorylated.
- 2) Again in reference to the label-free analysis, a more detailed methods section of how this analysis was run and quantified is necessary. E.g., clarifying whether spectral counting or precursor signal intensity was used in quantification, what was done with ambiguous peptides etc.
- 3) The authors describe sampling the heart ventricle. What size of sample was taken and what cell types were included? Different cell types can have different proteomic profiles, which could affect the outcome of the proteomics.
- 4) In the label free phospho-proteomics sampling, was a general proteomics sample also run? The lack of the phosphorylated peptide could be from lack of phosphorylation or the lack of protein. This should be controlled by the proteomics data set, as was done in the TMT-MS analysis. Without this, it is hard to tell the reason phosphosites are missing. It is noted on page 10 that statistical changes in these protein levels do not account for the difference in phospho-peptides, but the analysis of the proteome is not discussed in the methods or the intro for the label free section.
- 5) In reference to the TMT-MS analysis, was the proteome used to normalize the phosphor-proteome? If so, what was the approach used?
- 6) Beyond the methodological issues, in the TgErbB2, when the AG-825 was added, there is a distinct difference in the phenotype. Again, what times were looked at in this experiment for the control and experimental conditions? Moreover, were identified phosphosites tracked and does this phenotype occur via changes to the identified phospho-proteomes? I.e., can a direct link be drawn between what was identified in the phosphor-proteomics to what is seen in vivo.

Reviewer #2 (Remarks to the Author):

The work is a directed step towards clarifying the role that tyrosine phosphorylation has on the pathogenesis of heart failure. The authors tackle this challenge using global mapping and quantification of tyrosine phosphorylation in two separate mouse models of cardiac hypertrophy (TgErbB2 and R403Q-aMyHC). They show that each TG mouse model has a distinct signature in the phosphorylation of tyrosine. They use pathway analyses to identify model specific differences and tandem mass tagging to focus on myofilament phosphorylation. Th results show and mass tagging confirmed targets associated with hypertrophic and dilated cardiomyopathy. Further bioinformatic approaches showed mouse model specific differences in kinase activity signatures and modules of

protein-protein interactions. The authors go to show the inhibition of ErbB2 (using AG-825) in the TgErbB2 mice reverses or prevents the development of cardiac pathogenesis. Overall the amount of information is impressive and the authors do a considerable amount of bioinformatic analyses to underscore the differences between the mouse models. The last set of in vivo studies are not well incorporated in the manuscript and why they were performed in reference to the preceding data is unclear. Additional in vivo studies to address/validate some of the bioinformatic findings, particularly in reference to the differences in the findings of each animal model are needed to strengthen the overall impact of the work. Other major concerns are the manuscript lacks key information, including incomplete figure legends (e.g., Figure 2E-2H, Figure 3G), lack of referencing figures in the results (e.g., Fig. 7F) and information on the mice and procedures used for the several of the studies presented in Figure 7 (age, dosing procedure, histochemistry, etc.). Clear descriptions for the statistical analyses for several experiments are also lacking (e.g., Figure 1 and Figure 7). The overall descriptive aspect of the work are impressive, but the lack of coordination among the bioinformatics and in in vivo data sets, as well as the incompleteness of the manuscript significantly lowers impact of the study.

Reviewer #3 (Remarks to the Author):

In this paper, the authors explore the proteome of two mice models of hypertrophic cardiomyopathy (HCM); TgErbB2 and R403Q-aMyHCTg. The authors used various proteomics analyses which yielded the conclusion that tyrosine phosphorylation in HCM plays an important regulatory role.

Although the paper is novel and uses advanced omics approaches to prove the hypothesis, the paper has a few problems with control groups, which makes the story incomplete. For example, in figure 1, the authors present data on global Tyrosine phosphorylation and c-Src activation only in TgErbB2 mice, but not in R403Q-aMyHC mice. The data using TMT approach was applied only on the TgErbB2 mice, but not to the R403Q-aMyHC mice. The data showing the inhibitory effects of AG-825 was tested only on TgErbB2 mice and not on R403Q-aMyHC mice. Furthermore, the last figure, using AG-825 to reduce cardiac myocyte disarray and preserves contractile function in TgErbB2 Mice, does not add anything novel or sufficient to the paper, as this approach is known from the literature. Altogether, I believe that the paper is novel and the proteomics analysis has the potential to advance the field, however the mentioned comments above, together with the minor comments below should be considered:

1. The section that describes the choice of the mouse models should not be in the results, as it does not present any data. This section belongs to the introduction.
2. Figure 1A- please provide housekeeping gene as a loading control for this immunoblot. Instead of coomassie blue.
3. The paper is too long, especially the results and discussion sections.
4. It is suggested that the paper will have some grammatical edit.

Reviewer #1

The work discussed is interesting and novel and presents a solid discovery from omics levels analyses of these heart disease models. However, I have a few questions that should be address, in general relating to the methodological approach of the authors.

Reply: We are pleased by this overall comment/appreciation and thankful for your well-posed insights/suggestions.

1) When sampling the data, what was the time of day at which the mice were sac'ed? Was it at the same time for all the mice and if so, what was that time? The proteome and phospho-proteome of mice are known to change over the day and so differences in the time of sampling can have a significant effect on the proteins that are phosphorylated.

Reply: Mice were euthanized in the morning around 9-10 am. The rationale beyond this choice was the following. The mice must be sacrificed simultaneously, and given the labor-intensive approach, we opted for the morning time. For example, for the label-free experiments, we processed five mice per genotype (Ntg, ErbB2, and MyHC-R403Q); therefore, we had to move forward with 15 animals in parallel (Fig 2 and Fig 3), a process that we have done in triplicate.

2) Again in reference to the label-free analysis, a more detailed methods section of how this analysis was run and quantified is necessary. E.g., clarifying whether spectral counting or precursor signal intensity was used in quantification, what was done with ambiguous peptides, etc.

Reply: Precursor signal intensity MS1 was used for quantification with MaxQuant Software (Germany). Following your suggestions, we used the flow-through of phospho-tyrosine immuno-enrichment to quantify the whole proteome with TMT 9-plex labeling; then, we normalized the label-free data (MS1) to TMT intensity signals. We removed single spectra or data in which one or more reporter ions are missing. The data were first set at a 1% peptide false discovery rate (FDR). Their intensity was Log2 transformed, and then a median sweep was applied. Essentially, we processed the data as previously reported in Foster et al. J. Proteome Res., 2016.

3) The authors describe sampling the heart ventricle. What size of sample was taken and what cell types were included? Different cell types can have different proteomic profiles, which could affect the outcome of the proteomics.

Reply: After removing the atria, the net ventricle weight varied from 150 to 200 mg for both Ntg and MyHC-R403Q. Conversely, for the ErbB2 mice, typically, we obtained 200-250 mg of the left ventricle due to the massive LV hypertrophy already present in the genotype. Our sampling of the LV tissues comprised all cell types: ventricular cardiomyocytes, fibroblasts, and vessels. All LV tissue was homogenized for the label-free proteomics; therefore, all cell types are included in our analysis. However, for the immunoprecipitation assay, we invariably used 30 mg of protein homogenate lysates pooled from five mice for each assay. Finally, we used myofilament-enriched fractions for the TMT quantitative experiments; hence, they were made mostly by the cardiomyocyte myofilaments proteins.

4) In the label free phospho-proteomics sampling, was a general proteomics sample also run? The lack of the phosphorylated peptide could be from lack of phosphorylation or the lack of protein. This should be controlled by the proteomics data set, as was done in the TMT-MS analysis. Without this, it is hard to tell the reason phosphosites are missing. It is noted on page 10 that statistical changes in these protein levels do not account for the difference in phospho-peptides, but the analysis of the proteome is not discussed in the methods or the intro for the label-free section.

Reply: We fully agree with your comment. Due to the stochastic nature of the shotgun proteomics, we know that the lack of a non-modified or phosphorylated peptide does not necessarily mean that the

peptide is not there. Thus, we initially did label-free phospho-proteomics as a funnel screening. To this end, we employed the MS1 Maxquant approach and intra- and inter-experimental data normalization, using median subtraction (median sweep). Moreover, following the additional Reviewer's suggestion, we conducted a TMT-MS analysis on the Flow-through of those samples. The new data acquired and normalized to TMT for the whole proteome are now displayed in Figures 2 and Figure 3. As the Reviewer can readily appreciate, the outcome is not dissimilar to what was previously found with the initial approach. But we appreciated the opportunity to validate the original findings through the methodology suggested by the Reviewer.

The revised manuscript now includes a reference to the whole proteome analysis for the global heart TMT flow through analysis (see the Supplementary Table 2; page 11, second to the last paragraph). Same for the complete proteome analysis of the sarcomere proteins TMT (Supplementary Table 1; page 13, last paragraph) and phospho-tyrosine and phospho-serine/threonine analysis.

5) In reference to the TMT-MS analysis, was the proteome used to normalize the phosphor-proteome? If so, what was the approach used?

Reply: Yes. The full proteome of the sarcomere proteins was used to normalize the phospho-proteome. More in detail, data from the peptide phosphorylation is matched to the intensity of the full protein TMT signal. We converted to the logarithm scale the signal of phosphorylated peptide and that of full protein, then subtracted the logarithm of full protein intensity from the logarithm of peptide intensity. $\log_2(\text{peptide} / \text{protein}) = \log_2(\text{peptide signal}) - \log_2(\text{full protein})$. The peptides that did not find a protein match in whole proteome were discarded.

6) Beyond the methodological issues, in the TgErbB2, when the AG-825 was added, there is a distinct difference in the phenotype. Again, what times were looked at in this experiment for the control and experimental conditions? Moreover, were identified phosphosites tracked and does this phenotype occur via changes to the identified phospho-proteomes? I.e., can a direct link be drawn between what was identified in the phosphor-proteomics to what is seen in vivo.

Reply: We performed more experiments to increase the number of animals and compare the phenotype. We injected animals in the morning and at night: one injection 12 hrs apart from the other, for 14 days. We performed an echocardiographic evaluation on the morning of day 13th of the treatment, sacrificing the animals on the morning of day 14th of the treatment. Although the improvement in myocyte disarray was still evident, the initial differences observed by echo were no longer significant when increasing the animal number. We preserved and processed the hearts for proteomics, using the whole heart lysate to perform global proteomics, TMT 13-plex full proteome, phospho-tyrosine proteomics, phospho-Serine/Threonine proteome. The groups were divided into Ntg-Vehicle ($n=3$), Ntg-AG-825 treatment ($n=3$), TgErbB2-Vehicle ($n=3$) and TgErbB2-AG-825 treatment ($n=4$).

We identified 6,614 proteins, 4,732 Ser/Thr phosphorylated peptides and 346 phosphorylated Tyrosine peptides. We found a few phospho-sites that link directly with the effects seen in vivo. They collectively belong to EGFR Pathway. KSEA showed that nine kinases were significantly downregulated during AG-825 treatment as compared to vehicle. All those kinases belong to the downstream signaling of EGFR Pathway. Thus, the data show us that AG-825 treatment was effective in blocking ErbB2 and EGFR pathways; however not enough to reverse cardiac hypertrophy or preserve cardiac contractility. Therefore, any pharmacological interventions must be performed early in the cardiac hypertrophy progression.

Reviewer #2:

"The work is a directed step towards clarifying the role that tyrosine phosphorylation has on the pathogenesis of heart failure."- "Overall the amount of information is impressive and the authors do a considerable amount of bioinformatic analyses to underscore the differences between the mouse models." – " The last set of in vivo studies are not well incorporated in the manuscript and why they were performed in reference to the preceding data is unclear."

Reply: We appreciate your kind words/consideration for our work/study. Concerning your last remark, we know from the previous work from our collaborator (Dr. K. Gabrielson, Sysa-Shah P PLoS One 2012, Sysa-Shah P Cardiovasc Res 2016) that the early administration of lapatinib, or of a more specific ErbB2 inhibitor, such as AG-825, can halt cardiac hypertrophy and reduce fibrosis. However, we did not know if cardiac hypertrophy or myocyte disarray could be reversed using one of these inhibitors later in life. We chose the AG-825 compound because more specific for the ErbB2 receptor and found that 14-days treatment with AG-825 did not affect cardiac hypertrophy; however, we observed a reduction in fibrosis and an improvement of myocyte disarray. This novel observation suggests ErbB2 activity is directly linked with cardiac disarray in the TgErbB2 mouse model and might benefit other cardiomyopathies.

1) Additional in vivo studies to address/validate some of the bioinformatic findings, particularly in reference to the differences in the findings of each animal model are needed to strengthen the overall impact of the work.

Reply: We agree with your suggestions. Accordingly, we did perform additional animals to confirm the findings, after more mice were treated with AG-825 we lost the differences in heart contractility reported in the initial echocardiography studies. However, these samples helped us to have enough tissue to perform global total proteome quantification by TMT-13plex, phospho-serine/Threonine and Phospho Tyrosine proteomics. Serine/Threonine phospho-proteomics validated bioinformatics findings. KSEA is based on peptides' intensity signals from Ser/Thr and Tyrosine Kinases. New Figure 8, shows the new proteomics data that validate previous findings and the central involvement of EGFR Pathway.

2) Other major concerns are the manuscript lacks key information, including incomplete figure legends (e.g., Figure 2E-2H, Figure 3G),

Reply: Thanks for pointing to these deficiencies. We believe we have now fixed all these issues. Thank you.

3) Lack of referencing figures in the results (e.g., Fig. 7F).

Reply: In the revised manuscript we reference the figures in the results section. Thank you.

4) Information on the mice and procedures used for the several of the studies presented in Figure 7 (age, dosing procedure, histochemistry, etc.).

Reply: In the revised manuscript, page 17, first paragraph, we define the dose of AG-825 1mg/kg for 14 days and Masson's trichrome staining for fibrosis determination on slides. Mice age is 6-9 months.

5) Clear descriptions for the statistical analyses for several experiments are also lacking (e.g., Figure 1 and Figure 7).

Reply: Thanks for pointing this out. In the current version, we have incorporated this relevant information. Please, see page 29, the second to the last paragraph for Figure 1, and the first paragraph of page 38 for Figure 7.

6) The overall descriptive aspect of the work are impressive, but the lack of coordination among the bioinformatics and in vivo data sets, as well as the incompleteness of the manuscript significantly lowers impact of the study.

Reply: Thank you for this posed concern. In the current revision, we strived to better coordinate and integrate the bioinformatics, validated by global phospho-proteomics (including Ser/Thr/Tyr) and KSEA (Fig. 8).

Reviewer #3

In this paper, the authors explore the proteome of two mice models of hypertrophic cardiomyopathy (HCM); TgErbB2 and R403Q-aMyHC. The authors used various proteomics analyses which yielded the conclusion that tyrosine phosphorylation in HCM plays an important regulatory role. Although the paper is novel and uses advanced omics approaches to prove the hypothesis, the paper has a few problems with control groups, which makes the story incomplete. For example, in figure 1, the authors present data on global Tyrosine phosphorylation and c-Src activation only in TgErbB2 mice, but not in R403Q-aMyHC mice.

Reply: We thank you for the appreciation of our work and pointing out the problems with the control group. We agree about that, and, in the current revised version, we have updated Figure 1, including data concerning R403-aMyHC mice and a comparison in terms of Tyrosine phosphorylation levels. Fig 1 now shows that tyrosine phosphorylation levels are decreased in TgErbB2 and R403-aMyHC mice. SRC expression is increased in R403-aMyHC mice, and pSRC shows a trend to increase in the same mouse model without reaching statistical significance.

- 1) The data using TMT approach was applied only on the TgErbB2 mice, but not to the R403Q-aMyHC mice. The data showing the inhibitory effects of AG-825 was tested only on TgErbB2 mice and not on R403Q-aMyHC mice.

Reply: This is another well-taken point. Accordingly, we extrapolated the protein expression from the Flow-through from pTyr immuno-enrichment and quantified the complete proteome with TMT 9-plex. The previously label-free data (MS1 quantification) were normalized to whole proteome data and analyzed with the same bioinformatic tools. Unfortunately, we were unable to rescue the colony of R403Q-aMyHC mice. The breeders did not expand the colony, and, in the middle of the Covid-19 pandemic, was impossible to recover this TG line for more experiments.

Notwithstanding, we performed another batch of TgErbB2 mice and NTG with AG-825 and vehicle treatments. We went one step further, determining the phospho-Serine/Threonine proteome to validate findings of disease pathways and, more importantly, KSEA. The newly acquired data show that EGFR Pathway is central to the development of Cardiac Hypertrophy in the TgErbB2 model.

- 2) Furthermore, the last figure, using AG-825 to reduce cardiac myocyte disarray and preserves contractile function in TgErbB2 Mice, does not add anything novel or sufficient to the paper, as this approach is known from the literature.

Reply: We respectfully disagree on this point. Indeed, to the best of our knowledge, early administration of lapatinib or AG-825 in TgErbB2 mice can block cardiac hypertrophy and fibrosis development. However, previous work did not explore the effects on cardiac myocyte disarray. Hence, we aimed to test whether a similar treatment – performed in adulthood - could halt or reverse hypertrophy or improve myocyte disarray. We did more mice with AG-825 or vehicle treatment, and we could not confirm the preservation of cardiac contractility. Yet, we observed improved myocyte disarray.

- 3) Altogether, I believe that the paper is novel and the proteomics analysis has the potential to advance the field, however the mentioned comments above, together with the minor comments below should be considered: The section that describes the choice of the mouse models should not be in the results, as it does not present any data. This section belongs to the Introduction.

Reply: We have done what you suggested. Please, see the new Introduction (page 5, second and third paragraph).

4) Figure 1A- please provide housekeeping gene as a loading control for this immunoblot. Instead of Coomassie Blue.

Reply: Done. Please see page 6 (first paragraph). Thank you.

5) The paper is too long, especially the results and discussion sections.

Reply: Thanks for pointing this out. We tried our best to shorten both the Results and Discussion sections.

6) It is suggested that the paper will have some grammatical edit.

Reply: We have scanned our paper to target grammar mistakes, such as punctuation errors, misused words, misspellings, and more. Thanks.

Reviewers' comments:

Reviewer #1 (Remarks to the Author):

The revised manuscript "Altered Tyrosine Phosphorylation of Cardiac Proteins Prompts Contractile Dysfunction in Hypertrophic Cardiomyopathy" has satisfied all my review comments, save the direct link between the phospho-proteome and the in vivo data. Though I don't think that this is essential, it would very much strengthen the paper. Perhaps more importantly, the notes from some of the other reviewers, particularly about organization and length, are still an issue and I strongly recommend that the authors address these issues before publication.

Reviewer #2 (Remarks to the Author):

The authors have addressed several of my concerns. There are several additional editorial and stylistic issues that this reviewer thinks need to be addressed.

The authors do not reference the Fig 4C, Fig 4D, Fig 5F, Fig 8E or Supplemental Table 3 in the main draft. Referencing this work will help the reader better understand the figures and data.

The authors need to correct their use of abbreviations. This can make the manuscript difficult to read and understand. Many abbreviations are defined multiple times (e.g., HCM), the abbreviation sometimes appears before it is defined (e.g., NTg), and sometimes a word will be abbreviated differently depending on the location in the draft (e.g., R403Q-alphaMyHC vs . R403Q-alphaMyHC tg).

The abbreviation list is incomplete. Please include all the abbreviations used so the reader can follow use this to help them understand the manuscript.

I strongly encourage the authors to edit the last sentence in the abstract. "Thus, altered tyrosine phosphorylation may play a regulatory role in cardiac hypertrophic models, suggesting that tyrosine kinase inhibitors could be used therapeutically to treat hypertrophic cardiomyopathy." This seems to be an overstatement/interpretation of the data based on the current data in this manuscript.

This reviewer recommends the authors continue to shorten the discussion (and the results section) so as to make the presentation even more clear and concise with the goal of appealing to a larger readership.

Reviewers' comments:

Reviewer #1 (Remarks to the Author):

The revised manuscript "Altered Tyrosine Phosphorylation of Cardiac Proteins Prompts Contractile Dysfunction in Hypertrophic Cardiomyopathy" has satisfied all my review comments, save the direct link between the phospho-proteome and the in vivo data. Though I don't think that this is essential, it would very much strengthen the paper. Perhaps more importantly, the notes from some of the other reviewers, particularly about organization and length, are still an issue and I strongly recommend that the authors address these issues before publication.

Reply: We are pleased by your appreciation of our efforts to satisfy all reviewer's comments. We are thankful for your well-posed suggestion. We have tried to shorten the length of the manuscript. If we had more insights about which parts of the manuscript could be omitted or go to supplemental data, it would be more practical to reduce the length or change the manuscript's organization.

Reviewer #2 (Remarks to the Author):

The authors have addressed several of my concerns. There are several additional editorial and stylistic issues that this reviewer thinks need to be addressed.

The authors do not reference the Fig 4C, Fig 4D, Fig 5F, Fig. 8E or Supplemental Table 3 in the main draft. Referencing this work will help the reader better understand the figures and data.

Reply: We are thankful for your editorial suggestions and for noticing a few remaining issues. For references to Fig. 4C and D, see page 13, top paragraph. For the correction of Fig. 8E, please see the top sentence of page 20. This reference was also fixed in the corresponding Figure legend. The reference to Supplemental Table 3 was included on page 20, middle paragraph.

The authors need to correct their use of abbreviations. This can make the manuscript difficult to read and understand. Many abbreviations are defined multiple times (e.g., HCM), the abbreviation sometimes appears before it is defined (e.g., NTg), and sometimes a word will be abbreviated differently depending on the location in the draft (e.g., R403Q-alphaMyHC vs . R403Q-alphaMyHC tg).

The abbreviation list is incomplete. Please include all the abbreviations used so the reader can follow use this to help them understand the manuscript.

Reply: Thanks for pointing this out. We have updated the abbreviation list in pages 2-3.

I strongly encourage the authors to edit the last sentence in the abstract. "Thus, altered tyrosine phosphorylation may play a regulatory role in cardiac hypertrophic models, suggesting that tyrosine kinase inhibitors could be used therapeutically to treat hypertrophic cardiomyopathy." This seems to be an overstatement/interpretation of the data based on the current data in this manuscript.

Reply: Thanks for the well-posed suggestion. We have removed the suggestion of using tyrosine kinase inhibitors to treat HCM. See page 4, last sentence.

This reviewer recommends the authors continue to shorten the discussion (and the results section) so as to make the presentation even more clear and concise with the goal of appealing to a larger readership.

Reply: We have also commented to Reviewer#1 that we have tried to shorten the length and organization. However, without directly indicating what parts of the manuscript could be omitted or go to supplemental data, it is very challenging to shorten the manuscript further.